# OST-Bench: Evaluating the Capabilities of MLLMs in Online Spatio-temporal Scene Understanding

**Jingli Lin**[1,2*]**, Chenming Zhu**[1,3*]**, Runsen Xu**[1,4]**, Xiaohan Mao**[1,2]**, Xihui Liu**[3]
**Tai Wang**[1†]**, Jiangmiao Pang**[1†]

[1]Shanghai AI Laboratory, [2]Shanghai Jiao Tong University,
[3]The University of Hong Kong, [4]The Chinese University of Hong Kong
[*]Equal contribution      [†]Co-corresponding

https://rbler1234.github.io/OSTBench.github.io/

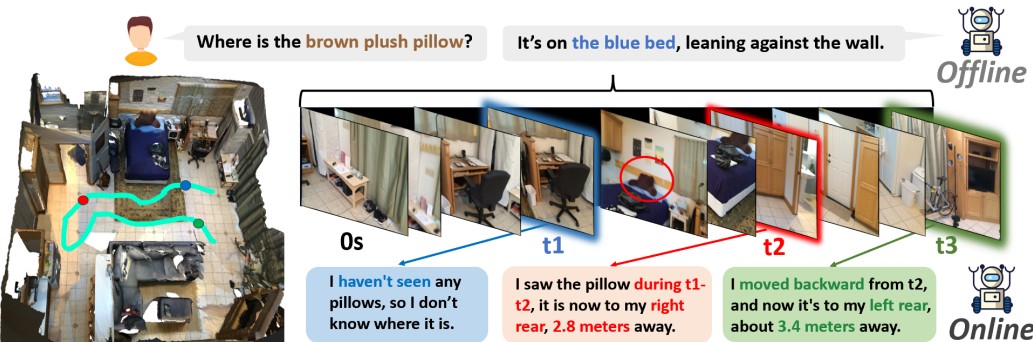

Figure 1: **OST-Bench** is designed from the perspective of an embodied agent dynamically exploring static indoor environments, with a focus on **online** and **spatio-temporal** understanding. Compared to the conventional offline setting (top right), which answers questions based on a fixed-length video of the scene, the bottom section illustrates our online setting: for the same question, the agent's answers evolve as it explores the scene, changing from blue (t1) to red (t2) to green (t3), reflecting its continuously updated understanding.

## Abstract

Recent advances in multimodal large language models (MLLMs) have shown remarkable capabilities in integrating vision and language for complex reasoning. While most existing benchmarks evaluate models under offline settings with a fixed set of pre-recorded inputs, we introduce OST-Bench, a benchmark designed to evaluate Online Spatio-Temporal understanding from the perspective of an agent actively exploring a scene. The "Online" aspect emphasizes the need to process and reason over incrementally acquired observations, while the "Spatio-Temporal" component requires integrating current visual inputs with historical memory to support dynamic spatial reasoning. OST-Bench better reflects the challenges of real-world embodied perception. Built on an efficient data collection pipeline, OST-Bench consists of 1.4k scenes and 10k question-answer pairs collected from ScanNet, Matterport3D, and ARKitScenes. We evaluate several leading MLLMs on OST-Bench and observe that they fall short on tasks requiring complex spatio-temporal reasoning. Under the online setting, their accuracy declines as the exploration horizon extends and the memory grows. Through further experimental analysis, we identify common error patterns across models and find that both complex clue-based spatial reasoning demands and long-term memory retrieval requirements significantly drop model performance along two separate axes, highlighting the

39th Conference on Neural Information Processing Systems (NeurIPS 2025) Track on Datasets and Benchmarks.

core challenges that must be addressed to improve online embodied reasoning. To foster further research and development in the field, our codes, dataset, and benchmark are available at `https://github.com/InternRobotics/OST-Bench`.

# 1 Introduction

In the real world, humans continuously perceive and update their understanding of the environment through sequential visual observations. At every moment, we are aware of our spatial state and how it evolves with respect to surrounding objects and scenes. We expect embodied agents to possess similar online scene understanding capabilities. For instance(Fig. 1), in an embodied navigation task[6, 52, 12, 28, 44], an agent should be able to incrementally construct a representation of its surroundings (*"I have seen a brown pillow on the bed in the bedroom."*), track its current status (*"I am now in the living room next to the bedroom, facing south."*), and reason about dynamic spatial relationships (*"The brown pillow is now on my rear left."*). Such awareness enables the agent to instantly respond to commands ( *"Go and get the brown pillow."*) and take correct actions.

Recent advances in multimodal large language models (MLLMs)[9, 18, 30, 39, 29, 49, 26] have shown remarkable capabilities in integrating vision and language for complex reasoning. However, most existing benchmarks [15, 31, 32, 8, 24, 5], evaluate models under **offline** settings, where reasoning is performed over a fixed set of pre-recorded inputs, such as reconstructed 3D scenes or images and videos, this does not capture the online nature of embodied tasks.

To address this gap, we introduce **OST-Bench**, a benchmark designed to evaluate **Online Spatio-Temporal understanding** from the perspective of an agent actively exploring a scene. The term *Online* emphasizes the agent's need to perceive, remember, and reason over incrementally received observations, rather than complete, pre-recorded scene data. The term *Spatio-Temporal* highlights the need to integrate current visual observation with historical memory to support dynamic spatial reasoning. To more accurately simulates real-world embodied perception, OST-Bench defines tasks from these perspectives: the **agent**, its surrounding **environment**, and their **relationship**, contains three main categories(Fig.2): (1) **Agent State**: the agent's understanding of its own state, (2) **Agent Visible Info**: the agent's dynamic interpretation of visible scene information, and (3)**Agent-object Spatial Relationship**: the agent's dynamic understanding of spatial relationships with objects, all posed in an online, temporally grounded fashion. OST-Bench comprises 1.4k real-world scenes sourced from the test and validation splits of ScanNet[20], Matterport3D[14], and ARKitScenes[11], accompanied by 10k QA pairs covering a diverse range of subtypes. Our benchmark provides a rigorous testbed for assessing the online spatio-temporal reasoning ability of MLLMs in realistic, embodied settings.

We evaluate leading MLLMs on OST-Bench and find its online spatio-temporal nature poses significant challenges for the models, even the most advanced models lag behind human performance by over 30%. Models perform poorly on tasks requiring complex spatio-temporal reasoning, with accuracy declining as exploration steps increase and memory grows under the online setting. Based on an in-depth experimental analysis, we observe a phenomenon which we term *Spatio-temporal Reasoning Shortcut*-when reasoning over long-term memory, models tend to avoid retrieving key information, instead taking shortcuts and relying on shallow, unsupported inferences; further, we design four tasks with different levels of difficulty to better delineate the models' capability limits, along both the spatial dimension (from single- to multi-step spatial reasoning) and the temporal dimension (from keyframe- to sequence-baesd context), and observe a clear performance drop on both dimensions. This reveals that both complex clue-based spatial reasoning and long-term memory retrieval are two distinct weaknesses that hinder the model's performance on OST-Bench, highlighting the core challenges that must be addressed to advance online embodied reasoning. Moreover, our fine-tuning analysis shows that data-driven training alone yields only limited improvement, suggesting that further progress will likely require advances in model architecture and training methodology rather than sheer data scaling.

# 2 Related Work

**Spatial Reasoning Benchmarks.** Early scene understanding benchmarks[8, 32, 24, 31, 50, 54] introduced diverse task taxonomies to comprehensively evaluate various aspects of visual scene interpretation, with spatial understanding consistently recognized as the most fundamental component.

| Dataset | Input Modality | Settings | Spatio-Temporal Awareness | Output Format | |
|---|---|---|---|---|---|
| | | | | Text | Num. |
| ScanQA [8] | Video/PC. | Offline | ✗ | ✓ | ✗ |
| SQA3D [32] | Video/PC. | Offline | ✗ | ✓ | ✗ |
| SceneVerse [24] | Video/PC. | Offline | ✗ | ✓ | ✗ |
| MMScan [31] | Video/PC. | Offline | ✗ | ✓ | ✗ |
| SpatialRGPT-Bench [19] | Image | Offline | ✗ | ✓ | ✓ |
| CV-Bench [38] | Image | Offline | ✗ | ✓ | ✓ |
| VSI [46] | Video | Offline | ✗ | ✓ | ✓ |
| **OST-Bench** | Video | Online | ✓ | ✓ | ✓ |

Table 1: **Comparison with other spatial reasoning datasets.** "PC." abbrev for "Point cloud". "Text" and "Num." represent whether the output is a string or a numerical value. Compared to other benchmarks, ours is clearly distinguished by its focus on the online setting and the requirement for spatio-temporal awareness in models.

Benchmarks such as ScanQA[8], SQA3D[32], SceneVerse[24], and MMScan[31] emphasized semantic understanding and incorporated object locations and spatial relations, they largely treated spatial relationships as semantic attributes, focusing primarily on complex spatial semantics rather than explicitly targeting spatial reasoning. With the rapid advancement of Multimodal Large Language Models (MLLMs), recent benchmarks have begun to place greater emphasis on spatial reasoning evaluation, SpatialR-GPT[19] and CV-Bench[38] require models to reason about 3D information, such as depth and distance from a single image, VSI[46] proposed a finer-grained categorization of spatial reasoning tasks, systematically evaluating models' ability to infer 3D scene layouts from 2D video inputs, covering both relative and absolute spatial relationships. Existing spatial reasoning benchmarks predominantly operate in an offline setting, focusing on static scenes and requiring models to perform reasoning over a fixed set of images or videos of predefined length. In contrast, our OST-Bench adopts an online setting, emphasizing dynamic scene understanding from an agent-centric perspective, and offers an alternative perspective for evaluating spatial reasoning capabilities. It includes a wider range of complex question types to assess more diverse and fine-grained spatio-temporal reasoning abilities.

**Video Benchmarks for Temporal Understanding.** Video benchmarks for temporal understanding require models to reason over both temporal and visual dimensions. Early efforts in video temporal understanding primarily focused on semantic comprehension from a third-person perspective[45, 47, 22, 43], mostly without considering 3D spatial perception. More recent benchmarks, driven by embodied task settings, have introduced characteristics such as: (1) egocentric perspective[23, 33, 13], where tasks are presented from a first-person viewpoint, (2) online inference[48, 16, 27, 16], requiring online processing of continuously streaming video input, and (3) spatial understanding[33, 27, 13], which evaluates models' awareness of spatial elements. However, spatial tasks in these benchmarks are often limited to 2D relationships or short-term motion cues, reflecting more of a content-level understanding rather than deeper spatial reasoning, lacking complex 3D spatial reasoning that requires integrating multi-view 2D observations into a coherent 3D representation. In contrast, OST-Bench is an egocentric, online temporal video benchmark that uniquely emphasizes 3D spatial reasoning, a core ability for real-world embodied tasks such as navigation and exploration.

## 3 OST-Bench

In this section, we present our comprehensive methodology for establishing OST-Bench, which comprises three core components: task formulation, the data collection and processing pipeline, and benchmark sample generation.

### 3.1 Task Formulation

Before introducing OST-Bench, we clarify the assumptions underlying our formulation of scene understanding. (1) While existing real-world datasets predominantly feature static scenes, we specifically focus on static environments in our current benchmark design, meaning the positions and states of objects remain unchanged during exploration; the agent is the only dynamic element. (2) There is no defined absolute coordinate system in the scene, so all spatial references are defined

relative to an anchor such as an object, viewpoint, or the agent itself. As a result, the position of any object or agent cannot be defined in isolation. All spatial measurements fall into four categories: relative distance, absolute distance, relative direction, and absolute direction.

A static scene consists of a set of immobile objects, and understanding such a scene involves reasoning about individual entities and their relationships[31], such as the **object attribute**(intrinsic properties of individual objects, including category, color, material, shape, size, and function), **the attribute / spatial relationship between objects**, and **the spatial relationship between objects and a given viewpoint** (provided either textually or via a virtual camera input). When a dynamic agent is introduced, it introduces new relational dynamics and opens up additional avenues for investigation. These can be categorized into three main categories that form the core focus of our benchmark evaluation: (1) **Agent state**: The position and orientation of the agent, which continuously change as the agent explores. (2) **Agent visible info**: The perceptual information available from the agent's point of view at a given moment includes the existence of visible objects, their count, diversity, and the timing of their appearance. The information visible to the agent is continuously updated as the agent explores the scene. (3) **Agent-object spatial relationship**: 3D spatial relations between the agent and objects, described by relative or absolute distance/direction, constantly change as the agent explores.

## 3.2   Meta-dataset Collection and Processing

**Base Dataset Acquisition.**  The three real scene datasets, ScanNet[20], ARKitScenes[11] and Matterport3D[14], contain rich scene information along with RGB-D videos/images and their corresponding camera information, totaling 7.6k scenes. Building on this foundation, EmbodiedScan[41] provided a large number of high-quality 9-DOF bounding box annotations for the objects in these scenes. MMScan[31] further enriched these scenes with a large number of highly quality, manually annotated object- and region-level semantic annotations. We selected a total of 1.4k scenes from the validation/test splits of these three datasets and constructed our dataset based on the annotations from EmbodiedScan and MMScan.

**Exploration Route Generation.** To construct an agent-centric exploration dataset, we require first-person videos of environments accompanied by camera parameters. While ScanNet and ARKitScenes provide such first-person videos along with camera pose data suitable for modeling agent trajectories, Matterport3D offers only multi-view images without continuous exploration paths. To address this limitation, we generate synthetic exploration trajectories within Matterport3D by constructing a graph of camera viewpoints and applying the minimum-spanning tree algorithm[36]. This ensures coherent movement and obstacle-free transitions between connected nodes. To maintain observation continuity, we enforce an image-overlap threshold between adjacent viewpoints. This approach enables us to simulate first-person exploration videos for Matterport3D scenes, complete with associated camera parameters.

**Visible Information Processing.** OST-Bench requires fine-grained visibility annotations at the frame level, which we define in two forms: attribute visibility and spatial visibility. Attribute visibility refers to the ability to determine the existence of an object based on a single frame. Even if an object is partially visible in a frame, as long as its visible portion is sufficiently large, you can infer attributes such as the object's type or color. Spatial visibility is used to generate questions in the OST-Bench that are related to the object's 3D spatial information. Therefore, for spatially visible objects, in addition to being attribute visible, we require that their center position, size, shape, and other spatial information can be inferred from observation. In practice, the attribute and spatial visibility of an object are determined by thresholding the projected area of its point cloud and the visibility of the vertices of its 9-DoF bounding box. Additional implementation details are provided in Appendix A.2.

## 3.3   Benchmark Samples Generation

**Rule-based Generation.** OST-Bench is designed in a multi-round dialogue format. In each round, the model receives a sequence of newly observed, temporally ordered frames, which are appended to all previously seen frames to simulate a streaming video input. At the end of the round, a new question is posed based on the accumulated observations. As the dialogue progresses, the input sequence grows incrementally, requiring the model to perform reasoning over an expanding spatio-temporal context. All questions are framed from an online perspective, grounded in the agent's current situation.

Our questions span three major categories: *Agent State, Agent Visible Info,* and *Agent–Object Spatial Relationships*. As illustrated in Fig.2, each main category contains multiple subtypes. Across all

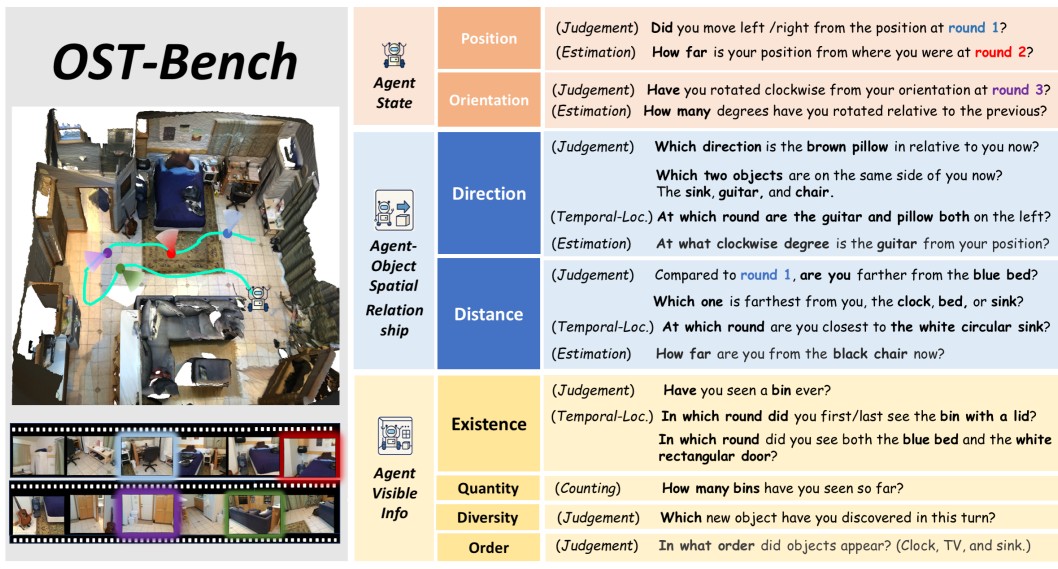

Figure 2: OST-Bench categorizes questions into three main categories. Each main category includes several subtypes; in total, the benchmark comprises 15 fine-grained question subtypes.

categories, the questions fall into four general formats: *Judgment*, *Counting*, *Temporal Localization*, and *Estimation*. Judgment questions evaluate the model's qualitative understanding of facts—whether something is true or not, or whether something has occurred; Counting questions assess the model's ability to quantitatively enumerate information; Temporal Localization questions test the model's ability to locate events along the time axis. (We use the round index as a discrete timestamp in OST-Bench); Estimation questions evaluate the model's ability to approximate measurable quantities, such as physical distances or angular differences.

Based on the processed meta-datasets, we define dedicated rule-based generation templates to construct corresponding data samples for each subtype within the three main categories. Detailed generation procedures for each fine-grained subtype, including rule definitions and templates used, are provided in Appendix A.3. Several representative samples of these subtypes are illustrated in Fig.2. Our benchmark comprises approximately 1.4k test and validation scenes selected from ScanNet, Matterport3D, and ARKitScenes. For each scene, we generate a single agent exploration trajectory. Along each trajectory, multiple dialogue rounds are defined, each containing a single question, resulting in a total of 10k questions across the dataset.

**Data Quality.** Ensuring high-quality benchmark data is crucial. Based on the high-quality manual annotations from Embodiedscan and MMScan, we design and iteratively refine tailored rule-based generation strategies for each subtask to ensure semantic validity, robustness, and clarity, avoiding common corner cases and ambiguities. To assess dataset quality, we employ a rigorous validation protocol in which questions are randomly sampled for manual review. Samples lacking sufficient information or containing incorrect answers are marked as invalid. Human evaluation results confirm that the dataset meets our strict quality standards, with an error rate below 5%, thereby ensuring a reliable and high-quality benchmark.

## 4 Experiments

### 4.1 Benchmark Models & Evaluation Metrics

We evaluate the performance of multiple multi-modal large language models (MLLMs), including both proprietary models (Claude-3.5-Sonnet[7], GPT-4o[34], GPT-4.1[35], Gemini-2.0-Flash[37], and its thinking variant) and open-source models (InternVL-2.5[17], QwenVL-2.5[10], LLaVA-Onevision[25], and LLaVA-Video[51] of different scales). Each model is tested in a zero-shot setting and conducts inference in a multi-turn dialogue format. (In addition to these general-purpose VLMs, we also include several models specifically designed with explicit spatial grounding or memory mechanisms to varying degrees, their results are reported in Appendix C.1.) To establish performance boundaries, we include two baselines: a human baseline and a chance-level baseline. For the

| Methods | Agent State | | | | Agent Visible Info | | | | | Agent-object Spatial Relationship | | | | | |
|---|---|---|---|---|---|---|---|---|---|---|---|---|---|---|---|
| | Position | | Orientation | | Existence | | Quantity | Diversity | Order | Direction | | | Distance | | |
| | JUD. | EST. | JUD. | EST. | JUD. | TEMP. | CNT. | JUD. | JUD. | JUD. | TEMP. | EST. | JUD. | TEMP. | EST. |
| *Proprietary* | | | | | | | | | | | | | | | |
| Claude-3.5-Sonnet | **65.1** | **36.2** | 50.6 | 30.3 | 85.8 | 67.0 | 57.9 | 57.1 | 60.0 | 39.2 | 18.3 | 21.9 | 43.8 | 54.9 | 19.0 |
| Gemimi-2.0-Flash | 59.7 | 27.3 | 56.7 | **36.5** | 89.6 | 70.8 | 59.4 | 78.7 | 55.6 | 42.0 | 17.8 | 21.5 | 45.5 | 48.9 | **27.1** |
| Gemimi-2.0-Flash(Thinking) | 57.4 | 36.3 | **61.1** | 33.4 | 88.1 | 74.8 | **61.9** | 63.6 | 73.1 | 50.9 | **51.7** | 23.3 | 51.1 | 56.8 | 22.7 |
| GPT-4o | 55.6 | 20.5 | 45.6 | 33.6 | 90.6 | 75.6 | 59.8 | 78.2 | 59.6 | 46.1 | 19.5 | 21.4 | 43.1 | 50.6 | 20.4 |
| GPT-4.1 | 64.2 | 30.7 | 60.8 | 33.2 | **90.8** | **78.0** | 60.6 | **82.1** | 70.8 | **51.5** | 23.4 | **28.6** | 44.9 | 53.6 | 23.9 |
| *Open-source* | | | | | | | | | | | | | | | |
| InternVL-2.5-8B | 51.7 | 26.1 | 49.4 | 40.4 | 86.3 | 51.3 | 56.4 | 60.7 | 38.4 | 37.2 | **33.8** | 22.8 | 43.0 | 42.9 | 27.9 |
| InternVL-2.5-38B | 56.7 | 31.8 | **54.6** | 38.4 | 91.7 | 74.7 | 61.1 | 79.8 | 62.1 | 42.1 | 20.6 | 27.7 | 42.7 | 42.5 | 28.1 |
| InternVL-2.5-78B | **60.8** | **34.4** | 49.9 | 40.7 | 90.7 | 74.4 | 65.9 | 77.9 | 61.2 | **43.4** | 22.4 | 17.8 | 46.7 | 44.4 | 22.9 |
| QwenVL-2.5-7B | 49.8 | 19.3 | 51.8 | 40.8 | 78.6 | 37.3 | 62.1 | 56.3 | 28.5 | 41.0 | 28.9 | 12.2 | 44.9 | 43.6 | 18.6 |
| QwenVL-2.5-32B | 51.0 | 31.1 | 53.5 | 39.4 | 85.3 | 64.8 | 59.2 | 73.4 | 41.8 | 39.5 | 24.9 | 25.7 | 43.6 | 39.1 | 20.3 |
| QwenVL-2.5-72B | 57.0 | 27.6 | 52.2 | 37.1 | 86.1 | 64.5 | 61.5 | 75.7 | 34.5 | 41.4 | 21.1 | 8.2 | 44.5 | 39.3 | 18.7 |
| LLaVA-Video-7B | 50.4 | 25.4 | 46.1 | 12.1 | 90.4 | 32.3 | 63.1 | 66.5 | 39.3 | 35.4 | 27.3 | 16.2 | 41.3 | 41.8 | 10.8 |
| LLaVA-Video-72B | 51.0 | 18.0 | 49.2 | **41.6** | 88.0 | 38.8 | 51.0 | 70.9 | 53.7 | 35.5 | 27.7 | 30.9 | 43.8 | 46.2 | 26.3 |
| LLaVA-Onevision-7B | 53.8 | 11.6 | 51.2 | 7.7 | 90.0 | 34.8 | **66.9** | 51.1 | 33.4 | 35.7 | 27.0 | **38.1** | 43.5 | 35.6 | 21.9 |
| LLaVA-Onevision-72B | 53.8 | 13.9 | 51.6 | 36.2 | 89.0 | 41.8 | 45.8 | 74.8 | 56.6 | 37.8 | 28.9 | 27.3 | 48.2 | 47.0 | 28.2 |
| *Baseline* | | | | | | | | | | | | | | | |
| Human-Level | 93.2 | 58.9 | 92.8 | 54.4 | 95.7 | 94.7 | 91.3 | 94.4 | 90.9 | 90.5 | 93.3 | 54.3 | 93.4 | 94.5 | 60.1 |
| Chance-Level | 50.0 | 37.8 | 50.0 | 39.3 | 50.0 | 29.1 | 25.0 | 33.0 | 25.0 | 36.0 | 33.2 | 47.6 | 36.0 | 31.2 | 30.3 |

Table 2: **Full evaluation results of OST-Bench.** This table reports the performance of each model across all fine-grained question subtypes, "JUD."/ "CNT." / "TEMP." / "EST." abbrev for "judgement","counting","temporal-localization", and "estimation".

| Methods | Avg | A. State | A. Info | AO. | JUD. | TEMP. | CNT. | EST. |
|---|---|---|---|---|---|---|---|---|
| *Proprietary* | | | | | | | | |
| Claude-3.5-Sonnet | 47.8 | 45.6 | 65.6 | 32.9 | 57.4 | 46.7 | 57.9 | 26.9 |
| Gemimi-2.0-Flash | 49.5 | 45.1 | 70.8 | 33.8 | 61.1 | 45.8 | 59.4 | 28.1 |
| Gemimi-2.0-Flash(Thinking) | 54.2 | 47.1 | 72.3 | **42.8** | 63.6 | **61.1** | **61.9** | 28.9 |
| GPT-4o | 48.7 | 38.8 | 72.8 | 33.5 | 59.8 | 48.6 | 59.8 | 24.0 |
| GPT-4.1 | 53.4 | **47.2** | **76.5** | 37.7 | **66.4** | 51.7 | 60.6 | **29.1** |
| *Open-source* | | | | | | | | |
| InternVL-2.5-8B | 44.6 | 41.9 | 58.6 | 34.6 | 52.4 | 42.7 | 56.4 | 29.3 |
| InternVL-2.5-38B | 50.8 | 45.4 | 73.9 | 34.0 | 61.4 | 45.9 | 61.1 | 31.5 |
| InternVL-2.5-78B | 51.1 | **46.5** | **74.0** | 32.9 | **61.5** | **47.1** | 65.9 | 29.0 |
| QwenVL-2.5-7B | 41.2 | 40.4 | 52.6 | 31.5 | 50.1 | 36.6 | 62.1 | 22.7 |
| QwenVL-2.5-32B | 46.9 | 43.8 | 64.9 | 32.2 | 55.4 | 42.9 | 59.2 | 29.1 |
| QwenVL-2.5-72B | 45.6 | 43.5 | 64.5 | 28.9 | 55.9 | 41.6 | 61.5 | 22.9 |
| LLaVA-Video-7B | 39.3 | 33.5 | 58.3 | 28.8 | 52.8 | 33.8 | 63.1 | 16.1 |
| LLaVA-Video-72B | 43.2 | 40.0 | 60.5 | **35.1** | 56.0 | 37.6 | 51.0 | 29.2 |
| LLaVA-Onevision-7B | 40.4 | 31.1 | 55.2 | 33.6 | 51.2 | 32.5 | **66.9** | 19.8 |
| LLaVA-Onevision-72B | 43.4 | 38.9 | 61.6 | 36.2 | 58.8 | 39.2 | 45.8 | 26.4 |
| *Baseline* | | | | | | | | |
| Human Level | 83.5 | 74.8 | 93.4 | 81.0 | 93.0 | 94.2 | 91.3 | 56.9 |
| Chance Level | 36.9 | 44.3 | 32.4 | 35.7 | 40.0 | 31.2 | 25.0 | 38.8 |

Table 3: **Model performance across main categories and question formats.** "A." abbrev for "Agent" and "AO." abbrev for "Agent-Object Spatial Relationship". The open-source and proprietary models with the highest and second-highest overall average scores are highlighted with bright green and light green marks.

human baseline, we ensure that participants have no prior exposure to the test scenes. The chance-level method adopts a random selection approach, randomly picks one answer from all possible choices for Judgement/Counting/Temporal-Localization questions, and for Estimation questions, it always outputs the mean value calculated from all potential numeric. As for the evaluation metrics, Judgement questions are considered correct if the model selects the same option as the ground truth. For Counting and Temporal Localization questions, the model's output—whether a number or a turn index—must exactly match the ground truth to be deemed correct. For Estimation questions, we adopt the Mean Relative Accuracy (MRA) metric from VSI[46] to score the similarity between the model's floating-point output and the ground truth.

## 4.2 Main Results

We report the performance of various models on our benchmark. Tab. 2 presents the performance of each model in all different subtype. Tab.3 summarizes the models' overall performance, including their overall average scores, average scores for each of the three main categories, and performance across different question formats. Additional model results and further analysis are provided in Appendix C.1.

**Substantial Gap between MLLMs' and Human's Performance.** Model accuracy lags significantly behind human performance, with consistent gaps across all question types as shown in Tab.2. According to Tab.3, even the most advanced models lag behind human performance by nearly 30% in the overall average score. This performance gap remains substantial across three main task categories and all question formats. These findings suggest that current MLLMs fall short on OST-Bench, illustrating how our benchmark presents a novel challenge, demanding stronger online spatio-temporal perception and reasoning capabilities. This observation motivates us to investigate further the reasons for the subpar performance of the models on this benchmark.

**Weak Spatio-Temporal Reasoning in MLLMs.** As shown in Tab. 2 and 3, a striking contrast can be observed across the three main task categories. Although most models achieve average scores close to 70% in the Agent Visible Info category, with performance in each subtype significantly above chance level, their scores in the Agent State and Agent-Object Spatial Relationship categories remain near chance level across all subtypes. This suggests that current models are capable of dynamically perceiving scene information with temporal awareness, but lack the ability to perform complex spatio-temporal reasoning.

**Performance Drop During Exploration.** As illustrated in Fig. 3, we observe a significant decline in model accuracy as the agent continues to explore with an increasing number of sequential observations in the online setting. This is expected: for each question, the agent must reason based on both the current observation and its historical memory. As the number of exploration turns grows, the amount of relevant past information the agent needs to retain also increases, naturally raising the difficulty of both perception and reasoning. We further analyze how performance evolves for two representative models, InternVL-2.5-38B and GPT-4.1, across the three main question categories. For Agent Visible Info questions, accuracy declines gradually and consistently over turns. In contrast, for Agent-Object Spatial Relationship and Agent State questions, performance drops sharply within the first few steps (typically within 2 to 4 turns) to near chance level, and remains low in subsequent turns.

**Comparison of Different Models.** When comparing the performance of different models in Tab. 2 and 3, we find that proprietary models demonstrate significantly stronger performance compared to open-source ones. For different variants of the same open-source model, scaling from smaller configurations (7B/8B) to larger ones (>32B) consistently leads to notable performance gains, particularly on the questions under the Agent visible info category; Enabling the "thinking" mode in Gemini-2.0-Flash results in substantial improvements over the original version, especially on the questions with Temporal Localization format and the those under the Agent-Object Spatial Relationship category. This suggests that the thinking mode effectively enhances both spatial and temporal awareness.

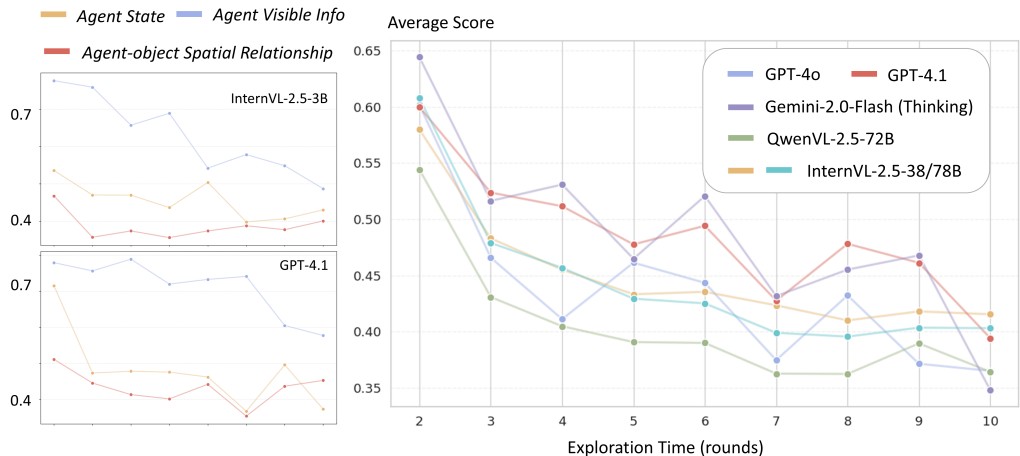

Figure 3: **Model performance over exploration time.** The right side shows a general decline in answer accuracy for all models; the left side illustrates the accuracy trends across three main categories for InternVL-2.5-38B and GPT-4.1.

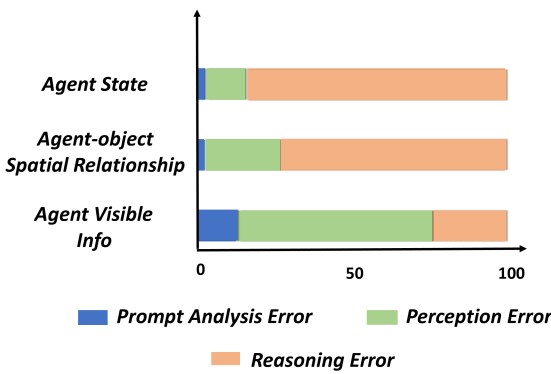

Figure 4: Distribution of three error types across the three task categories in OST-Bench.

Figure 5: An example of Spatio-temporal Reasoning Shortcut, the green text indicates correct reasoning by the model, while the red text highlights wrong reasoning.

## 4.3 Experiment Analysis

### 4.3.1 Insights from Model Explanations

To gain deeper insight into the weaknesses of models on OST-Bench, we prompt them to output not only their final answers but also their reasons. We manually examine the model outputs and identify the sources of errors. Since the model's inference process involves three key stages: understanding and following prompts, extracting information from observations, and performing spatio-temporal reasoning. Based on the stage at which the failure occurs, we categorize errors into three types: (1) **Prompt Analysis Error**, arising from the model's failure to correctly interpret the task setup or follow the given instructions; (2) **Perception Error**, where the model fails to accurately extract information from the visual observations by overlooking or misidentifying objects; (3) **Reasoning Error**, caused by incorrect spatio-temporal reasoning based on the information perceived. These three error types exhibit a clear progressive relationship. We select several representative open-source/proprietary models (GPT-4o, Gemini-2.0-Flash-Thinking, InternVL-2.5-78B) and examine 30 error cases for each major category per model, totaling 270 manual in-depth inspections.

**Error Distribution Statistics on OST-Bench.** The statistical results in Fig.4 show that Prompt Analysis Errors are relatively rare across all three major task categories, indicating that models generally understand the novel tasks and instructions introduced by OST-Bench. Perception Errors are the dominant failure mode for the Agent Visible Info category. In contrast, for tasks requiring more complex spatio-temporal reasoning, such as Agent–Object Spatial Relationships and Agent State, Reasoning Errors constitute a substantial portion of the failures. Based on the number of errors per task category and their distribution across the three error types, we estimate that Reasoning Errors account for over 60% of all errors, making them the primary bottleneck limiting current MLLM performance on OST-Bench.

**Spatio-temporal Reasoning Shortcut of MLLMs.** OST-Bench requires models to reason online over space and time, leveraging past observations to build spatial connections between the current state and prior states or previously seen objects. Within our in-depth error analysis, the model's Reasoning Error reflects a lack of this ability and reveals a common phenomenon as follows: The model tends to take shortcuts in reasoning, performing shallow and unsupported inference based on minimal information, and is reluctant to retrieve and utilize key information from long-term memory that could aid in answering the question. We name this phenomenon as *Spatio-temporal Reasoning Shortcut*. As shown in Fig. 5 example, the model correctly identifies that a television appeared in earlier frames and recognizes its own positional change over time. However, it makes an unfounded inference that the TV must now be behind it based solely on the fact that the TV is currently not visible, without using available spatial anchors such as the locations of a table or chair that could help establish a grounded reference frame. Additional examples of such shortcut behaviors are provided in Appendix C.2 to further illustrate their prevalence.

### 4.3.2 Cross-View Analysis

While most models struggle with complex spatio-temporal reasoning over sequentially growing memory, we introduce a targeted subset of OST-Bench to better delineate the capability boundaries of

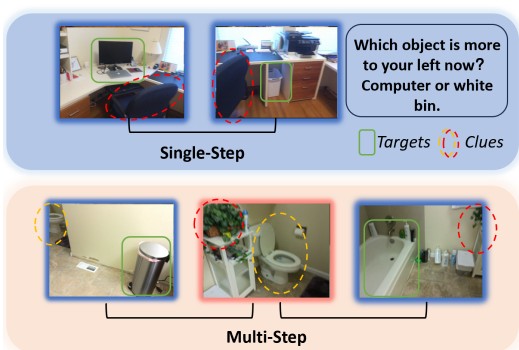

Figure 6: Single- vs. multi-step spatial connection settings. Target objects and spatial clues are highlighted.

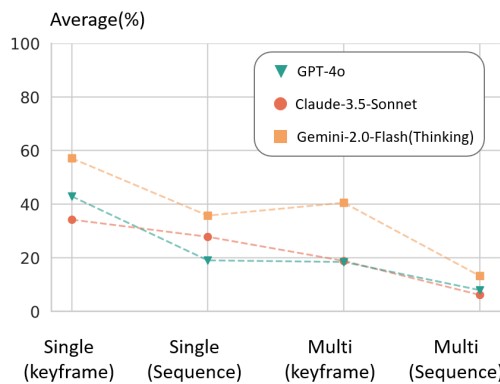

Figure 7: Model performance across four task settings: keyframe- vs. sequence-based context, and single- vs. multi-step spatial connection.

models. Questions of this subset focus on the spatial relationship between the agent and two objects that appear in different frames (e.g., *"Which object is more to your left?"*). It requires the model to construct cross-view spatial connections to answer correctly. It evaluates performance across two dimensions:

(1) **Single- vs. Multi-step Spatial Connection.** In the single-step setting, the spatial connection between two target objects can be directly inferred by analyzing the pair of the two frames that contain them. In contrast, the multi-step setting demands higher-level reasoning capabilities, where single-step pairwise frame analysis proves insufficient. This scenario requires the model to integrate spatial cues across multiple keyframes (typically more than two), iteratively establishing pairwise relationships between frames to enable chained reasoning through intermediate steps. As illustrated in Fig. 6, in the single-step case, the spatial connection between the computer and the white bin can be directly inferred through the shared objects (chair and table) in the single pair of images. While in the multi-step case, establishing the spatial connection between the bathtub and the gray trash bin necessitates an anchor image to bridge intermediate objects (trash bin → toilet → potted plant → bathtub), forming a multi-step spatial reasoning chain.

(2) **Keyframe- vs. Sequence-based Context.** In the keyframe-based setting, all keyframes that contain target objects or spatial cues sufficient to solve the problem are directly provided as input. In contrast, the sequence-based setting embeds these keyframes within a longer memory sequence that includes many irrelevant frames. The model must identify and leverage the relevant ones, thereby testing its capacity for long-term memory retrieval and reasoning.

This subset provides an opportunity to examine the model's performance across different levels of difficulty. We construct this dataset using a hybrid approach of rule-based generation and manual filtering. We curate 200 questions and evaluate three advanced MLLMs: Gemini-2.0-Flash (Thinking), GPT-4o, and Claude-3.5-Sonnet. For each question, models are required to provide both the answer and its reason. Only when both the final answer and the reason are correct is the response counted as correct. Based on our evaluation, as the results shown in Fig.7, we report the following key findings: (1) As tasks change from single to multi-step spatial connecting setting, which requires more complex reasoning, all models experience a substantial drop in accuracy; (2) Long-memory challenges further degrade performance. When models are required to locate relevant frames from sequence-based input rather than keyframe provided directly, accuracy drops significantly. In the most challenging tasks, which need to establish multi-step spatial connection in sequence-based context, all models fall to around 10% accuracy. The results show that the model's performance drops significantly when faced with either complex clue-based spatial reasoning requirements or long-term memory retrieval demands. OST-Bench exemplifies this dual challenge, as it requires models to retrieve information from a long, temporally extended memory while simultaneously constructing spatial relationships by integrating cues from multiple images to perform multi-step reasoning. These two factors jointly contribute to the poor performance observed on OST-Bench, highlighting the need to advance both capabilities in future model development.

### 4.3.3 Fine-tuning Analysis

To dig deeper into the upper bound of current models' capabilities and to better understand how much of the performance gap can be recovered through training with in-domain data, we conducted fine-tuning experiments on several representative models. Following a procedure similar to that used for constructing the benchmark samples, we generated training data from 7k training scenes across ScanNet, Matterport3D, and ARKitScenes, yielding a total of 50k annotated samples. All models were fine-tuned for a single epoch, and the results are summarized in Tab. 4. Overall, all evaluated models achieved performance gains exceeding 10% after fine-tuning. However, a closer analysis reveals several important insights:

| Method | Setting | Overall | JUD. | EST. | CNT. | Temp-Loc. | A State | A Info | AO |
|---|---|---|---|---|---|---|---|---|---|
| QwenVL2.5-7B | Zero-Shot | 41.2 | 50.1 | 22.7 | 62.1 | 36.6 | 40.4 | 52.6 | 31.5 |
| | Fine-Tuned | 54.0 | 59.0 | 41.2 | 74.6 | 50.2 | 48.3 | 69.8 | 43.5 |
| InternVL2.5-8B | Zero-Shot | 44.6 | 52.4 | 29.3 | 56.4 | 49.2 | 41.9 | 58.6 | 34.6 |
| | Fine-Tuned | 57.4 | 64.1 | 38.5 | 74.9 | 57.5 | 44.0 | 79.3 | 46.3 |
| InternVL2.5-38B | Zero-Shot | 50.8 | 61.4 | 31.5 | 61.1 | 45.9 | 45.4 | 73.9 | 34.0 |
| | Fine-Tuned | 60.2 | 68.4 | 44.1 | 73.1 | 56.1 | 50.8 | 81.7 | 47.5 |

Table 4: **Performance comparison of models under zero-shot and fine-tuned settings.** "A." abbrev for "Agent" and "AO." abbrev for "Agent-Object Spatial Relationship".

- Among the three major task categories, the largest improvements emerged in Agent Visible Info tasks — particularly for models with smaller parameter sizes. In contrast, the other two task categories, despite showing some gains, remained at or below 50% accuracy. This indicates that even with in-domain adaptation, models still struggle with tasks that demand complex spatio-temporal reasoning. Simple supervised fine-tuning on OST-Bench is therefore insufficient to resolve its core challenges.

- All four question formats benefited from fine-tuning, yet deeper inspection of predictions reveals more nuanced observations. Although fine-tuning improves the scores of Estimation (EST) and Judgement (JUD) tasks, closer inspection reveals that these gains do not reflect genuine reasoning improvements: models frequently output nearly identical values or default to the same option across samples, indicating reliance on dataset-specific shortcuts or memorization rather than true understanding. Moreover, their instruction-following ability degrades post-finetuning, with many responses failing to provide both the final answer and the required reasoning.

While fine-tuning significantly improves raw performance, a considerable gap remains compared to human-level accuracy. This highlights two key points: (1) data-only supervised fine-tuning is insufficient to solve the challenges posed by OST-Bench — improvements may also be required on the model architecture or training methodology side; and (2) the benchmark itself is both challenging and robust. Despite being constructed using templates, OST-Bench resists shortcut learning and cannot be easily exploited through superficial patterns in the training distribution.

## 5 Limitations and Conclusion

In this work, we propose OST-Bench, a novel benchmark for evaluating the online spatio-temporal reasoning capabilities of MLLMs. By emphasizing both online processing and spatio-temporal understanding, OST-Bench more accurately reflects the complexities of real-world perception and reasoning. Our extensive evaluation of leading MLLMs shows that OST-Bench poses significant challenges for models, particularly in tasks requiring complex spatio-temporal reasoning and maintaining answer accuracy as input accumulates over time in an online setting. We hope the public release of OST-Bench will serve as a catalyst for future research in online embodied understanding. We assume that the environment remains static. However, in real-world scenarios, object states and positions often change due to interactions with humans or agents. Additionally, our benchmark focuses solely on the agent's online perception and reasoning abilities, capturing only one aspect of real-world embodied tasks. Other crucial capabilities, such as interactive behaviors and active manipulation, are not considered in our current setting. These limitations highlight promising directions for future research and benchmark development.

# 6 Acknowledgement

This work is funded in part by the National Key R&D Program of China, and Shanghai Artificial Intelligence Laboratory.

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

# Appendix

## A   Benchmark Details

This section provides additional details on the construction of our benchmark, including the algorithm used for route generation, the method for determining object visibility, the rules for benchmark sample generation, and summary statistics of the generated data.

### A.1   Exploration Route Generation

While ScanNet and ARKitScenes offer egocentric video sequences with associated per-frame camera parameters, Matterport3D provides, for each scene, n camera positions distributed throughout the environment. From each position, k images are captured at different viewing angles, as illustrated in Fig. 8. We aim to leverage this information to construct a simulated trajectory of an agent exploring the scene from a first-person perspective. As mentioned in the main paper, the trajectory must satisfy two key requirements: (a) **Path continuity**, the movement between adjacent frames should be smooth, avoiding abrupt spatial jumps over short time intervals. (b) **Observation continuity**, adjacent frames in the video must have a certain degree of visual overlap, which is crucial for providing the cross-frame visual continuity necessary for constructing a coherent 3D understanding of the scene. The videos provided by ScanNet and ARKitScenes naturally satisfy both of these requirements.

The video we aim to generate is a sequence of tuples $\{(n_i, k_i, c_i)\}$, where $n_i$ denotes the camera position index among the n predefined locations, $k_i$ indicates the viewing angle index among the k available viewing angles at that position, and $c_i$ is the corresponding captured image. Based on the two aforementioned requirements(Fig.8), (a) We first construct a minimum spanning tree(MST) $T(N, E)$ over all camera positions using Prim's algorithm, where edge weights are defined by the

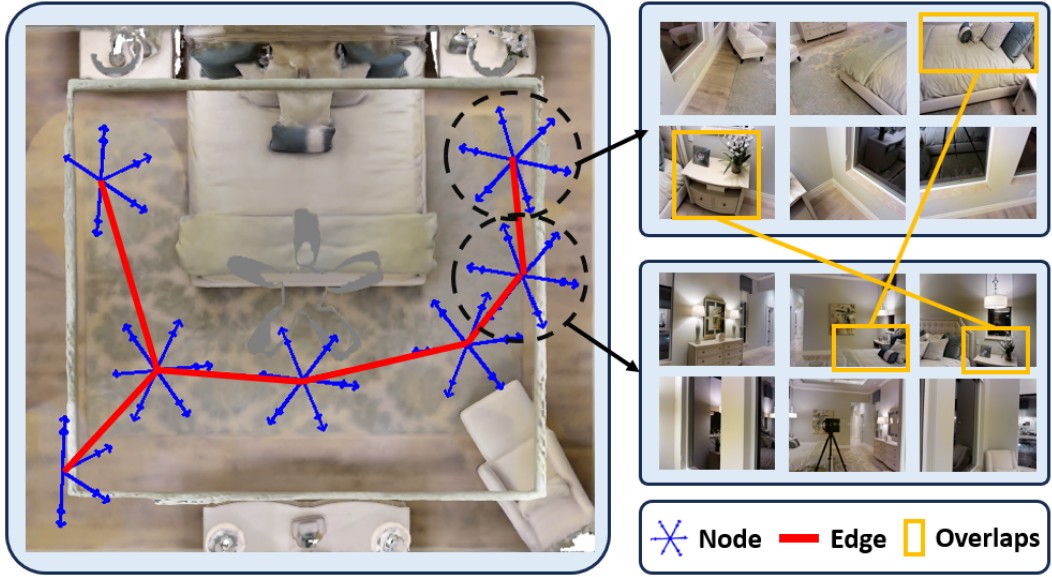

Figure 8: **Illustration of the route generation process.** The radial arrows represent multiple different viewing angles at a position, and the red edges denote connections generated by the MST algorithm. The right part shows the captured images for each viewing angle of two adjacent nodes. The agent can only move along the edges of the tree, and adjacent frames are required to have a certain amount of overlap.

Euclidean distances between positions. We constrain the agent's movement to either transitions between neighboring positions connected by edges in the MST ($E_{n_i, n_j} \in E$), or changes in viewing angles at the same position. This design ensures path continuity throughout the simulated trajectory. (b) We enforce that adjacent images in the sequence must have sufficient visual overlap. That is, for any $i \geq 0$, the overlap between images $c_i, c_{i+1}$ must satisfy $Overlap(c_i, c_{i+1}) > threshold$. This constraint preserves observation continuity across frames. Based on these two rules, we perform a random walk over the nodes to generate the sequence. Starting from a randomly selected initial state with a random tuple $(n_0, k_0, c_0)$, at each step, we randomly select a valid and previously unseen tuple representing the next state and append it to the sequence. This process continues until no valid tuples remain or the sequence reaches a predefined length.

It is important to note that the generated videos ensure continuity in terms of paths and observations, but do not guarantee temporal continuity (i.e., they only provide discrete frame ordering without information on the time intervals between frames). However, since our benchmark setting uses rounds as discrete timestamps, such temporal information is not required, and the provided data is sufficient for our purposes.

## A.2 Visible Information Processing

**Attribute Visibility.** For the attribute visibility of objects, to reduce computational complexity, we first apply a necessary condition: if an object is visible, then at least one of its 3D points must be projectable onto the 2D image plane within the image boundaries and without occlusion. This condition allows for the rapid elimination of most invisible objects in each image. For objects satisfying this condition, we project the surface points of their bounding boxes onto the image's 2D plane. We first compute the projected area $A_2$ without considering occlusion or image boundaries. Then, we calculate the visible area $A_1$ by accounting for occlusions and restricting projections to within the image bounds. An object is deemed visible if either (1) the ratio of visible to total projected area, $A_1/A_2$, exceeds a predefined threshold, or (2) the absolute visible area $A_1$ is sufficiently large.

**Spatial Visibility.** For the spatial visibility of objects, building on attribute visibility, we further check whether at least five vertices of the object's 9-DoF bounding box are visible in the frames observed so far. If this condition is met, we assume the object's center position, size (length, width, height), and related spatial information are all available, thus satisfying the criteria for spatial visibility.

## A.3   Rule-based Generation

Our OST-Bench comprises three major categories: *Agent State*, *Agent Visible Info*, and *Agent-object Spatial Relationship*. Within these categories, we define a total of 15 question subtypes. Data samples are generated through a rule-based approach, guided by a set of principles outlined below.

(1) **Multi-round Dialogue Format.** OST-Bench adopts a multi-round dialogue setup. In each round, 4–5 new frames from the video are selected sequentially in chronological order as new observations and appended to the historical observation sequence. Each question is asked at the timestamp of the last frame in the current round. All information after this timestamp is considered unavailable, and we ensure that the question is answerable based solely on the observations up to that timestamp.

(2) **Sample Pool Construction and Selection.** For each question subtype, we exhaustively generate all possible data samples to form a candidate pool. We ensure that no identical question-answer pair appears across different dialogue rounds (although the same question might occur, the answers must differ). In each round, we first randomly select a question subtype and then randomly select a data sample from its corresponding candidate pool as the question for that round.

(3) **Object Reference.** Object references in questions are divided into two types. The first is category-level reference, where a category word is used to refer to all instances of that category (e.g., *"How many books are there in the room?"*). The second is instance-level reference, where a specific grounded description is used to uniquely identify a single object.(e.g., *"Where is the yellow-covered book labeled with the word 'atomic'?"*). These descriptions are sourced from MMScan's object-level annotations. To eliminate ambiguity, we ensure that this referred object is the only instance of its category within historical observations.

(4) **Memory-based Reasoning Requirement.** To rigorously test a model's ability to reason over long-term memory and avoid overly simple questions, we ensure that no question can be answered using only the newly added observations in the current round. Each question requires integrating information from both the current and previous dialogue rounds. For example, we ensure that at least one relevant object is absent from the observations in the current round, thereby requiring the model to recall it from prior rounds.

(5) **Ensuring Clarity and Avoiding Ambiguity.** To ensure the validity and clarity of the questions and to avoid controversial or ambiguous cases, we impose specific thresholds during sample generation so that the answers are unambiguous and clearly inferable. For example, when a question involves comparing two distances, we require the difference between the distances to exceed a predefined threshold to ensure a significant contrast. Similarly, for questions such as determining whether an object is on the left or right, we require the object to be clearly positioned on one side. Objects located near the decision boundary (e.g., close to the center) are excluded to prevent ambiguity in interpretation.

Fig.9 presents the predefined templates used for generating questions across different subtypes. The specific generation strategies for each subtype are detailed below:

**Agent State.** This category encompasses tasks that require the agent to judge or estimate its own spatial state, including its position and orientation. Since there is no globally defined coordinate system in OST-Bench, all measurements are made relative to a specific historical time point.

- *Position (Judgement)*: In this type, the task is to determine whether the agent has moved to the left or right (forward or backward), relative to its position and orientation at the end of a previous round $T_1$. The question is formulated as a binary choice, with the correct answer being either *left* or *right*(*forward* or *backward*). Let $P_1$ and $O_1$ denote the position and orientation at the end of round $T_1$, and $P_2$ denote the current position. We compute the parallel and perpendicular components of the vector $P_2 - P_1$ with respect to $O_1$. A question is generated only if the absolute value of either component exceeds a predefined threshold (1 meter). The correct answer is determined by the sign of the respective component: a positive value indicates forward or right, while a negative value indicates backward or left.

- *Position (Estimation)*: In this subtype, the task is to estimate how far the agent has moved from its position at the end of a previous round $T_1$. The ground-truth answer is defined as the Euclidean distance between the agent's current position and its position at $T_1$.

| | | |
|---|---|---|
| **Agent State** | *Position(JUD.)* | **Q:** Assuming the direction at the end of {round ID} is forward, did you move a certain distance left or right / forward or backward from that position?
**O:** [left, right] / [forward, backward] |
| | *Position(EST.)* | **Q:** How far is your current position from where you were at the end of {round ID}? (in meters) |
| | *Orientation(JUD.)* | **Q:** Using your orientation at the end of {round ID} as a reference, has your current orientation rotated clockwise or counterclockwise by a certain angle (<180) relative to that orientation?
**O:** [clockwise, counterclockwise] |
| | *Orientation(EST.)* | **Q:** Using your orientation at the end of {round ID} as a reference, how many degrees has your current orientation rotated clockwise/counterclockwise relative to the previous orientation? |
| **Agent–object Spatial** | *Distance(JUD.)* | **Q1:** {object1}, {object2}, and {object3}, which one is the closest to/farthest from you now?
**O1:** [{object1}, {object2}, {object3}]
**Q2:** Compared to the end of {round ID}, are you now closer or farther away from {object}?
**O2:** [closer, farther]
**Q3:** Compared to the end of {round ID}, are you now closer to or farther away from {object1}/{object2}? |
| | *Distance(TEMP.)* | **Q:** In which round were you closest to/farthest from {object}? |
| | *Distance(EST.)* | **Q:** Please recall {object}, what is the horizontal distance between you and this object now (in meters)? |
| | *Distance(JUD.)* | **Q1:** Is the {object} to your left/right now?
**O1:** [left, right]
**Q2:** Which direction is {object} to you now: front left, front right, rear left, or rear right?
**O2:** [front left, front right, rear left, rear right]
**Q3:** Which two objects are on the same side of you now? {object1}, {object2}, and {object3}. |
| | *Distance(TEMP.)* | **Q:** At the end of which round were both of {object1} and {object2} on your left side? |
| | *Distance(EST.)* | **Q:** Based on your current orientation, in what (counter)clockwise direction (in degrees) is {object} from your position? |
| **Agent Visible Info** | *Existence(JUD.)* | **Q:** Remember, have you seen any {object type} so far?
**O:** [Yes, No] |
| | *Existence(TEMP.)* | **Q1:** When did you first discover/last see {object} (index of the turn)?
**Q2:** In which round did you see both {object1} and {object2} simultaneously? |
| | *Quantity(CNT.)* | **Q:** Remember, how many {object type}(s) have you seen so far? |
| | *Diversity(JUD.)* | **Q:** Which one was newly discovered in this round, {object1}, {object2} or {object3}?
**O:** [{object1}, {object2}, {object3}] |
| | *Order(JUD.)* | **Q:** What will be the first-time appearance order of {object type1}, {object type2} and {object type3}?
**O:** [{order1}, {order2}, {order3}, {order4}] |

Figure 9: **Rule-based generation templates for all subtypes in OST-Bench.** Placeholders to be filled with specific content are marked in red, and question focal points are highlighted in blue. "JUD."/ "CNT." / "TEMP." / "EST." are abbreviations for "judgement","counting","temporal-localization", and "estimation"; "Q" and "O" denote "Question" and "Options"

- *Orientation (Judgement)*: This binary-choice question asks whether the agent has rotated clockwise or counterclockwise by an angle(less than 180 degrees) relative to its orientation at the end of round $T_1$. We compute the angle between the current orientation vector and the one at the end of $T_1$. To exclude ambiguous borderline cases, questions are generated only if the angle lies within the intervals $[\theta, 180 - \theta]$ or $[180 + \theta, 360 - \theta]$, where $\theta$ is a threshold used to exclude borderline cases. Angles within the first interval indicate clockwise rotation, while those within the second indicate counterclockwise rotation.

- *Orientation (Estimation)*: In this question type, the task is to estimate how many degrees the agent has rotated, clockwise or counterclockwise, relative to its orientation at the end of a previous round $T_1$. The answer is given as the angle between the current orientation and the orientation at the end of round $T_1$.

**Agent Visible Info.** All objects involved in this category of questions must satisfy the attribute visibility constraint, meaning that their existence must be identifiable from past observations. This category evaluates the model's understanding of agent visible information, including subtasks such as object existence, quantity, diversity, and the order of appearances.

- *Existence (Judgement)*: This type asks whether a certain category was visible in any of the previous observations. The answer is binary: *yes* or *no*. To balance positive and negative

samples, we generate questions for object categories that do not appear in prior observations with a 50% probability.

- *Existence (Temporal Localization)*: This type includes two forms of queries: (1) Identifying the earliest/latest round in which a specific object was visible; (2) Identifying the round in which two specific objects were simultaneously visible. For both forms of queries, we ensure the answer is unique—i.e., there is exactly one round that satisfies the condition.

- *Quantity (Counting)*: This task requires counting how many objects of a specified category were visible in past observations. To avoid trivial cases, we exclude questions where the correct answer is one. Additionally, to balance the distribution, negative samples—where the target category does not appear at all—are introduced to constitute 25% of the total samples.

- *Diversity (Judgement)*: This question type asks which object is newly observed in the current round. The agent must choose one object from three candidates, all of which are visible in the current observation. Among them, only one has not appeared in any previous round, while the other two have been seen before.

- *Order (Judgement)*: This question type involves determining the appearance order of three different object categories. The agent must select the correct sequence from four given permutations. We ensure that the first appearance round of each object category is distinct to avoid ambiguity in ordering.

**Agent-Object Spatial Relationship.** This category focuses on constructing spatial metric relationships between the agent and a specific object $O$ at a specific time $T$. The distance between the agent and object $O$ at time $T$ is defined as the shortest distance from the camera coordinate to any point in the object's point cloud. The angle of object $O$ relative to the agent at time $T$ is computed as the angle between the camera's horizontal orientation vector and the vector pointing from the camera to the center of object $O$. All objects involved in this category must satisfy the spatial visibility constraint, which means that their center coordinates, dimensions (length, width, height), and other spatial properties must be reliably obtainable from previous observations.

- *Distance (Judgement)*: This question type includes three forms of queries: (1) determining which of the three objects is currently farthest from or closest to the agent; (2) judging whether the current distance between the agent and a specific object is greater or smaller than the distance at the end of a previous round; (3) judging whether the current distances between the agent and two specific objects are greater or smaller than those at the end of a previous round, with four possible answer choices. For the first form, at least one object must be invisible in the current round, and the distance to the correct answer object must differ significantly (i.e., by more than a predefined threshold) from the distances to the other two objects. For the second and third forms, the change in distance between the two time points must also exceed the threshold to ensure a meaningful distinction.

- *Distance (Temporal Localization)*: This task asks the agent to identify the round in which it was closest to or farthest from a specific object. The distance in the correct round must be significantly smaller (for closest) or larger (for farthest) than in all other rounds.

- *Distance (Estimation)*: This query requires estimating the current distance between the agent and a specific object, which is invisible in the current round and thus requires recalling information from previous rounds.

- *Direction (Judgement)*: This question type includes three forms of queries: (1) judging whether a specific object is currently on the agent's left or right side; (2) judging whether a specific object currently lies in the left-front, left-back, right-front, or right-back quadrant relative to the agent; (3) identifying which two out of three objects are currently on the same side of the agent. For the first two forms, we enforce angular thresholds by excluding objects whose relative angles fall within 10 degrees of the decision boundaries between sides or quadrants, thereby avoiding ambiguity. For the third form, at least two of the three objects are invisible in the current round, forcing the model to rely on memory.

- *Direction (Temporal Localization)*: This query asks the agent to identify the round in which both objects A and B were located on the same side (left or right) relative to the agent. We ensure that in each round, both objects are clearly on either the left or right side (at least 10

degrees away from the decision boundary), and that there is exactly one round satisfying this condition.

- *Direction (Estimation)*: This query requires estimating the angle, clockwise or counterclockwise, of a specific object relative to the agent's current orientation. The object is not visible in the current round, requiring retrieval from prior observations.

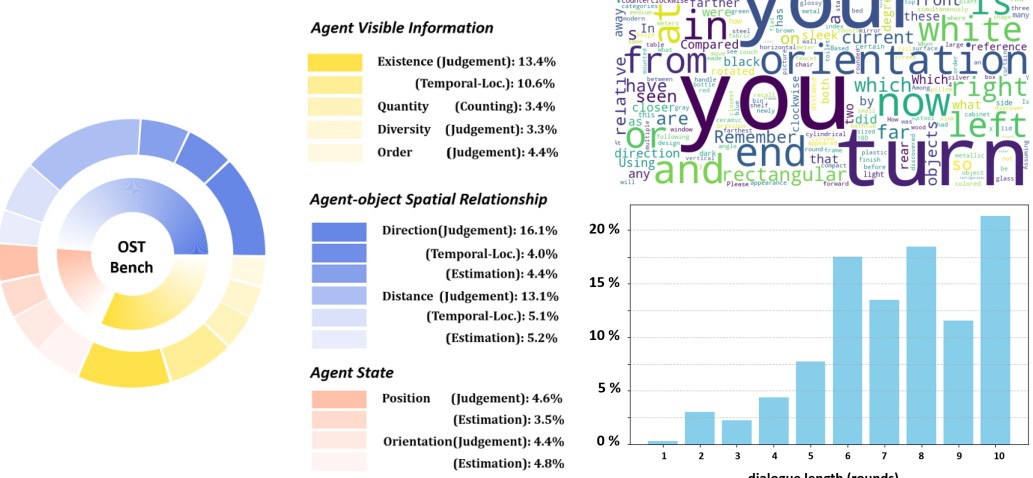

Figure 10: Distribution of sample counts across different subtypes in OST-Bench.

Figure 11: Word cloud (top) and dialogue length distribution (bottom) of OST-Bench.

## A.4 Statistics

Based on the generation methods described above, OST-Bench totally consists of 1.4k trajectories(a trajectory per scene) and 10k data samples. The distribution of sample counts across different subtypes is shown in Fig. 10. We also present in Fig. 11 the word frequency distribution in OST-Bench (visualized as a word cloud), as well as the distribution of dialogue lengths.

## A.5 Benchmark Examples

In Fig. 16 and 17 we provide more examples from our benchmark, including a total of 12 data samples from two scenes (exploration trajectories).

## B Implementation Details

For the multi-round dialogue, we first provide a system prompt to inform the models of the task setup. In each round, we sequentially input a set of images representing new video frames, along with a prompt containing a question, as illustrated in Fig.12. For judgment questions, we include the options in the prompt. For the other three qusetion formats (estimation, counting, and temporal-localization), we prompt the model to output a specific numerical value and explicitly instruct it to answer the question. This instruction is necessary, as we observed during experiments that models may otherwise refuse to respond, claiming insufficient information.

For proprietary models, we interact with the OpenAI and Anthropic APIs, both of which support multi-round dialogue with image inputs. In these APIs, models are invoked by explicitly specifying their model names. For the OpenAI API, we use *gpt-4o* for GPT-4o, *gpt-4.1* for GPT-4.1, *gemini-2.0-flash* for Gemini-2.0-Flash, and *gemini-2.0-flash-thinking-exp* for its thinking variant. For the Anthropic API, we use *claude-3-5-sonnet-latest* to access Claude-3.5-Sonnet. The system prompt is set to the task description, and each round's input includes newly added images and questions. For open-source models (InternVL, QwenVL, LLaVA-Onevision, and LLaVA-Video), we manually construct the multi-round context by concatenating the dialogue history, new images, and the current prompt as the input at each round. To avoid out-of-memory errors, input images are resized accordingly. For

| | | |
|---|---|---|
| **System Prompt** | | "Assume you are currently exploring a room where all objects are stationary. Over time, you change your position and orientation within the room and take images.
Now, I will engage you in a multi-round dialogue (a total of {num of rounds} ). In each round, I will provide you with {num of images per round} images taken from the beginning to the end of that round. Please answer my questions based on your state(position/orientation) at each round's end (last image)." |
| **User message** | | **\<image\>** + "For the {round ID}, these are the {num of images per round} images in chronological order. The question for this turn is: {question}. To answer this question, you need to combine information from past rounds. Please give me your answer and reason in a JSON format." |
| | *Judgement* | Please choose the answer from {options} . |
| | *Counting/ Temporal-Loc/ Estimation* | Please provide a numerical value as the result. The information I provided is sufficient for you to infer the value; do not refuse to answer! |

Figure 12: Model input content, including the system prompt and inputs for each round. Text placeholders to be filled are highlighted in red, while the green <image> token represent image placeholders to be filled.

models with up to 8 billion parameters, inference is run on a single NVIDIA A100 GPU. For models with 32 billion parameters or more, we perform multi-GPU inference using 8 NVIDIA A100 GPUs via model and data parallelism. Additionally, we implement multithreaded processing to accelerate the inference of open-source models.

## C   Experiment Analysis Details

### C.1   Spatially-Grounded Model Evaluation

In addition to general-purpose VLMs, we further evaluate several representative models that incorporate spatial grounding and memory mechanisms to varying degrees, including Spatial-MLLM[42], VLM-3R[21], and LLaVA-3D[53].

- Spatial-MLLM and VLM-3R follow a VGGT[40] + VLM architecture and take RGB image sequences as input, where VGGT provides geometry-aware scene representations.
- LLaVA-3D leverages RGB-D image sequences to encode 3D spatial information into 2D token embeddings.

All three models were trained on spatial reasoning datasets and achieved strong results on their respective benchmarks. We evaluate their performance on OST-Bench and compare them to their corresponding base models — Spatial-MLLM vs. QwenVL2.5-3B, and VLM-3R / LLaVA-3D vs. LLaVA-Video-7B.(Tab 5) The key findings are summarized below:

- **Only VLM-3R delivers consistent gains over its base model.** Spatial-MLLM and LLaVA-3D exhibit substantial performance drops, whereas VLM-3R shows steady improvements, particularly in Agent State, Agent-Object Spatial Relationship, and Estimation tasks.
- **Instruction-following ability degrades noticeably.** Compared to their base models—which reliably follow prompts and output both answers and reasoning—all three grounded models struggle to adhere to the required response format. Spatial-MLLM is constrained to producing only floating-point values or multiple-choice options, while all three models frequently omit reasoning or generate incoherent explanations.
- **Limited generalization beyond training-aligned distributions.** Despite excelling on spatial reasoning datasets such as VSI and MMScan, Spatial-MLLM and LLaVA-3D fail to generalize effectively to OST-Bench, which features more diverse and temporally grounded prompts. VLM-3R demonstrates partial transferability, yet its gains remain modest.

These observations suggest that while memory-enhanced spatial grounding can improve performance on tasks aligned with model pretraining objectives, it often comes at the expense of generalization. Such models may lose part of the base LLM's robustness — underperforming on previously simple tasks (e.g., Agent Visible Info or Counting questions in OST-Bench) and struggling with instruction-following in out-of-distribution settings. They tend to excel only on in-domain tasks and transfer poorly to broader benchmarks like OST-Bench, which involve more diverse and complex reasoning demands.

| Method | Overall | JUD. | EST. | CNT. | TEMP. | A. State | A. Info | AO. |
|---|---|---|---|---|---|---|---|---|
| (base) QwenVL2.5-3B | 34.8 | 47.9 | 18.7 | 59.4 | 19.8 | 34.2 | 47.5 | 25.7 |
| Spatial-MLLM | 26.8 | 37.3 | 21.9 | 29.5 | 15.3 | 25.5 | 39.4 | 20.9 |
| (base) LLaVA-Video-7B | 39.3 | 52.8 | 16.1 | 63.1 | 33.8 | 33.5 | 58.3 | 28.8 |
| VLM-3R | 42.9 | 55.1 | 28.3 | 49.6 | 36.0 | 39.9 | 58.1 | 34.4 |
| LLaVA-3D | 30.1 | 46.1 | 5.9 | 13.5 | 36.3 | 29.7 | 38.4 | 26.3 |

Table 5: **Performance comparison between specially designed models and their corresponding base models.** "A." abbrev for "Agent" and "AO." abbrev for "Agent-Object Spatial Relationship".

## C.2 More Findings in Tables

**Difficulty of Estimation Tasks.** As shown in Table 2 in the main paper, models perform particularly poorly on estimation tasks, achieving scores well below the chance-level baseline. Humans also struggle with these questions, obtaining significantly lower scores compared to other task categories. This is because estimation questions go beyond innate human perceptual abilities. Humans are better at perceiving spatial relationships approximately than estimating spatial measurements precisely, requiring not only spatial reasoning but also extensive empirical knowledge accumulated from experience.

**Detection Success vs. Counting Failure.** As shown in Table 2 in the main paper, models achieve notably high scores on object-existence questions, demonstrating a strong ability to identify whether and when objects appear. However, their performance drops significantly for object-quantity tasks, which require counting. Upon examining specific cases, we found that models frequently confuse whether objects across frames are the same or distinct, mistaking two different objects as identical or failing to track the same object across frames. This suggests that the task demands not just detection capabilities but also cross-frame reasoning.

**The Illusion of Better Distance Understanding.** As shown in Table 2 in the main paper, models appear to perform slightly better on Agent-object distance questions compared to Agent-object direction, but this advantage is superficial. This is primarily due to the *Spatio-temporal Reasoning Shortcut* phenomenon: models tend to assume that objects currently visible are closer, while those out of view are farther away, without engaging in genuine spatial reasoning. Although this heuristic can occasionally lead to correct answers, since such patterns do occur in a small portion of our benchmark, it fails to generalize. As a result, models still perform poorly on Agent-object distance questions overall.

## C.3 Cases of Three Error Types

In Fig.13, we present examples of the three types of errors: *Prompt Analysis Error*, *Perception Error*, and *Reasoning Error*. In the first example of Prompt Analysis Error, the prompt explicitly requires the model to output a specific quantity. However, the model fails to interpret this requirement correctly and responds with "no" instead of providing a numerical answer such as "0". In the second example, the model misunderstands the meaning of the word "discover" in the prompt. It assumes that partially seeing the keyboard in Round 1 does not count as a discovery and that only fully observing it in Round 2 qualifies as such. This misinterpretation leads to an incorrect answer. In the two Perception Error examples, the model fails to correctly identify a washbin located in the corner of the room and only detects one of the two lamps in the bedroom, missing the other. In the Reasoning Error examples, although the model correctly understands the prompt and accurately perceives the location of the target objects, it makes an error in reasoning about their spatial relation with the agent, leading to incorrect conclusions.

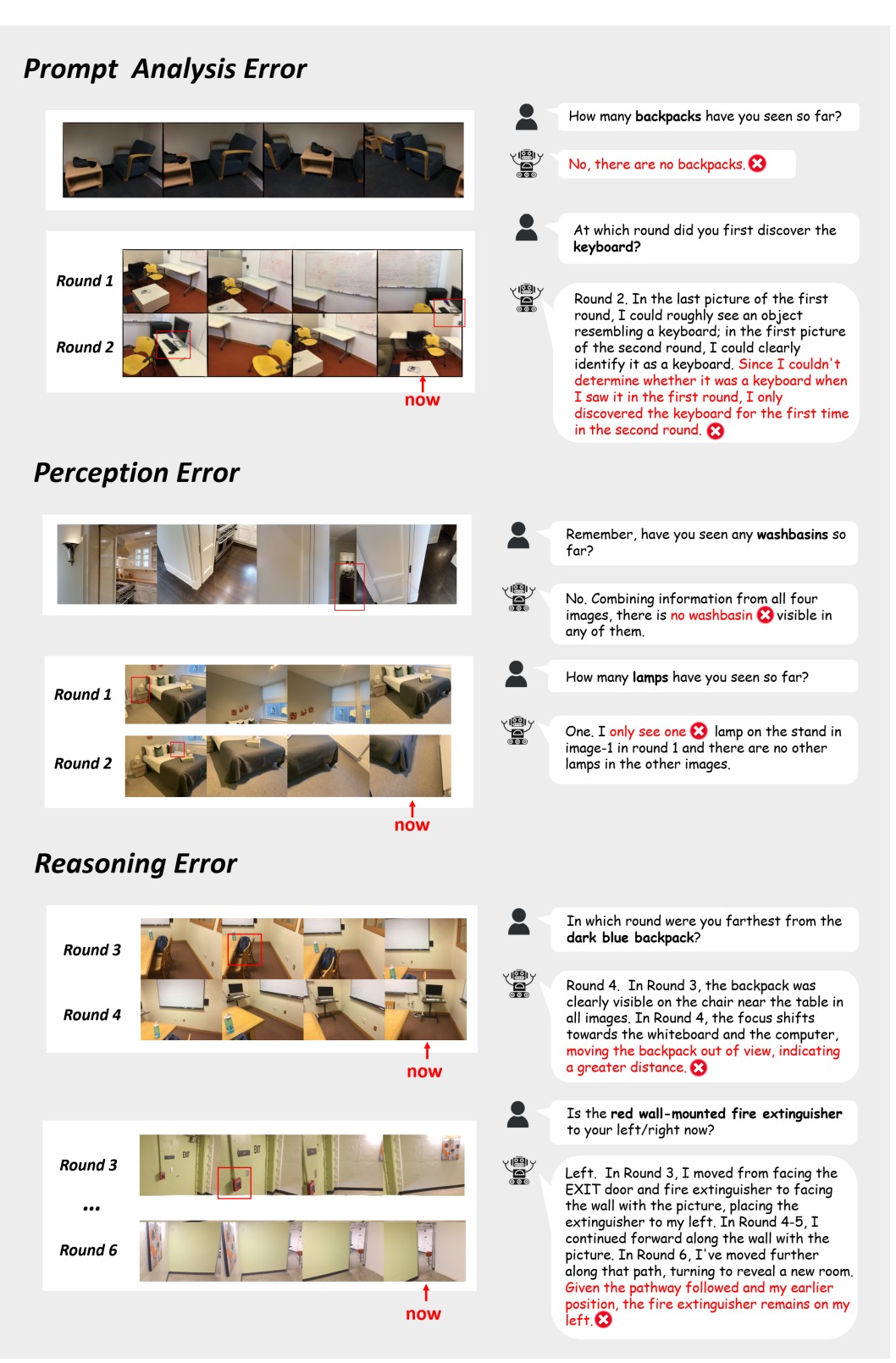

Figure 13: Illustrative Examples of the Three Error Types: Prompt Analysis Error, Perception Error, and Reasoning Error.

## C.4 Cases of Spatio-temporal Reasoning Shortcut

In the main paper, we have discussed the *Spatio-temporal Reasoning Shortcut* phenomenon exhibited by the models. In Fig.14, we provide additional examples to further demonstrate the prevalence of this behavior. For clarity, we display only the key video frames relevant to each question. Temporal expressions in the questions and model responses are replaced with t1, t2, and t3, and marked above the corresponding frames. All of these examples demonstrate the model's tendency to rely on shortcuts in spatio-temporal reasoning.

In the first example, GPT-4o incorrectly infers that the blackboard has moved closer simply based on its transition from being invisible to visible, ignoring spatial cues such as the chairs and the decorations on the wall. In the second example, Gemini-2.0-Flash performs a seemingly correct inference using only two frames (the current and target frames), concluding that the wall currently in front of the agent is adjacent and perpendicular to the wall in t1, while disregarding intermediate frames that contain crucial contradictory evidence. In the third example, InternVL-2.5-78B observes that the TV was on the right side of the room in earlier frames and then directly assumes it remains there when it becomes invisible. In the fourth and fifth examples, the models make incorrect judgments due to the target object being invisible in the specific frames. In the sixth example, the model only focuses on the frames where the stand appears and the current frame, while skipping over intermediate frames that indicate the agent turned around, wrongly assuming that the current orientation is aligned with the previous one.

## C.5 Subset Construction Process for Cross-View Analysis

As mentioned in the main paper, when constructing the dataset for the Cross-View subset, we first generate an initial batch of data using a rule-based method and then manually filter the data to obtain the final set of 200 samples. Our rule-based construction method for generating the Cross-view subset with different levels of difficulty is described as follows:

(a) **Single-Step Spatial Connection.** We first iterate over all possible object pairs $(O_1, O_2)$ in the scene. For each object pair, we traverse all possible frame pairs $(F_1, F_2)$ within the video sequence. A frame pair is selected if it satisfies the following conditions: (1) $O_1$ is visible in $F_1$ but not in $F_2$; (2) $O_2$ is visible in $F_2$ but not in $F_1$; (3) $F_1$ and $F_2$ share at least one overlapping object. This setup ensures that the spatial relationship between $O_1$ and $O_2$ can be inferred via single-step reasoning. All tuples $(O_1, O_2, F_1, F_2)$ satisfying these constraints are collected as initial data for the **keyframe-based context**. To construct the **sequence-based context**, we embed $F_1$ and $F_2$ into a video sequence $V$ that includes frames not containing $O_1$ or $O_2$, resulting in tuples of the form $(O_1, O_2, V)$.

(b) **Multi-Step Spatial Connection.** Similarly, we iterate over all object pairs $(O_1, O_2)$ and traverse all frame triplets $(F_1, F_2, F_3)$ from the video sequence. A triplet is selected if it meets the following conditions: (1) $O_1$ is visible in $F_1$ but not in $F_2$ or $F_3$; (2)$O_2$ is visible in $F_2$ but not in $F_1$ or $F_3$;(3) $F_1$ or $F_3$ share at least one overlapping object;(4) $F_2$ or $F_3$ share at least one overlapping object;(5) $F_1$ and $F_2$ have no overlapping objects. This configuration ensures that solving the problem requires multi-step reasoning. All valid tuples $(O_1, O_2, F_1, F_2, F_3)$ satisfying these constraints are collected as initial data for the **keyframe-based context**. Similarly, to construct the **sequence-based context**, we embed $F_1$, $F_2$ and $F_3$ into a video sequence $V$ that includes frames not containing $O_1$ or $O_2$, resulting in tuples of the form $(O_1, O_2, V)$.

# D  Inference Time of the Models

Although OST-Bench does not impose real-time constraints, we conducted a supplementary study on models' inference time, indirectly reflecting the delay in decision-making exhibited by the models in real-world embodied tasks. Since the inference time of proprietary models is also affected by network latency, we restrict our analysis to open-source models and report their inference time per question.

The Fig.15 illustrates how the model's inference time per question changes as the duration of exploration increases. The results reveal a clear trend: as exploration time increases and more historical context accumulates, inference latency grows rapidly. When the number of dialogue rounds becomes large (e.g., beyond 10), the inference time becomes prohibitively high, especially for large-scale models, making real-time interaction impractical. This latency surge stems from the fact that any frame in history may contain critical information, forcing the model to attend to a growing

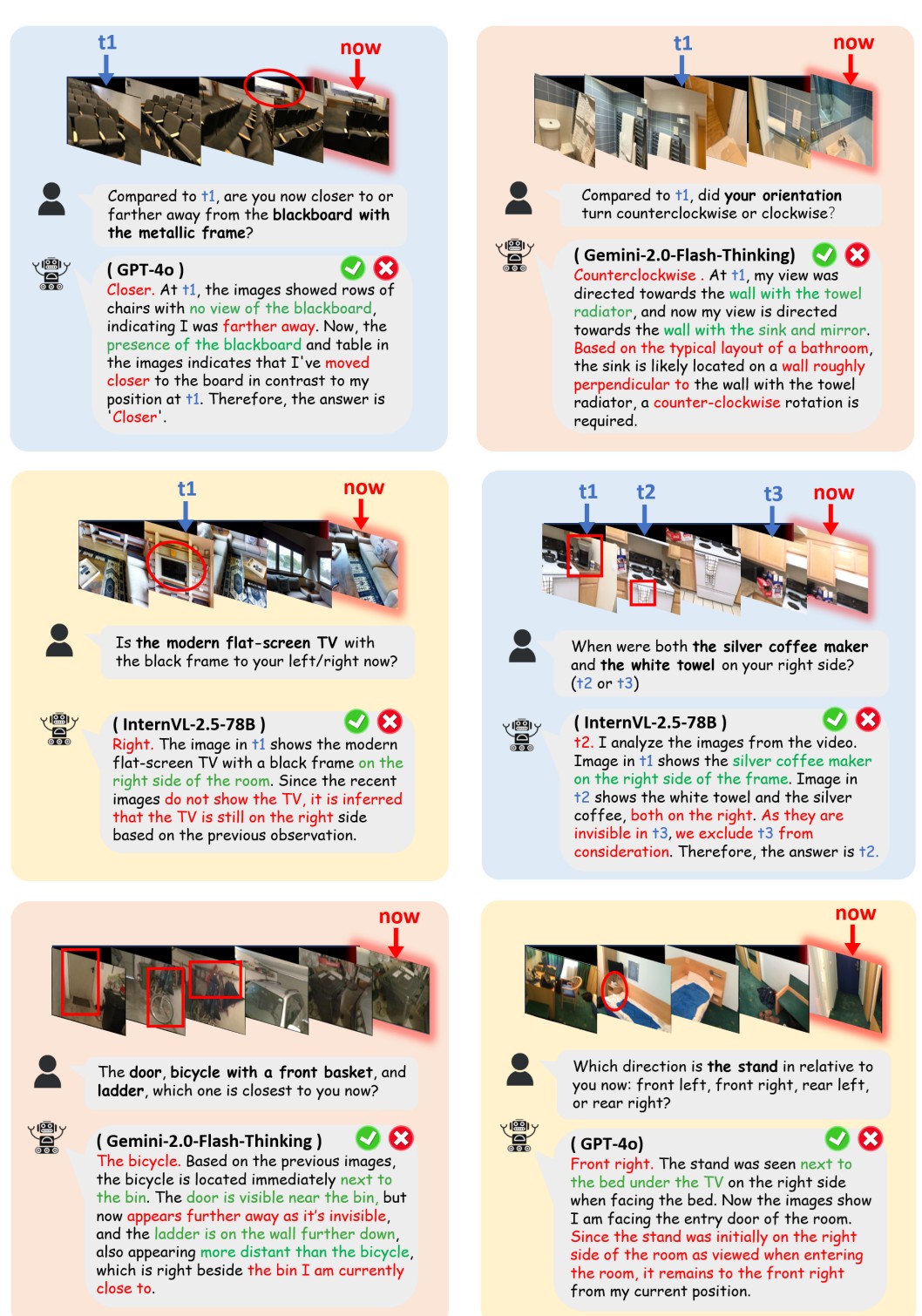

Figure 14: **More examples of Spatio-temporal Reasoning Shortcuts.** Green text marks correct reasoning; red indicates errors. For clarity, only key video frames relevant to each question are shown, with temporal references replaced by t1, t2, and t3.

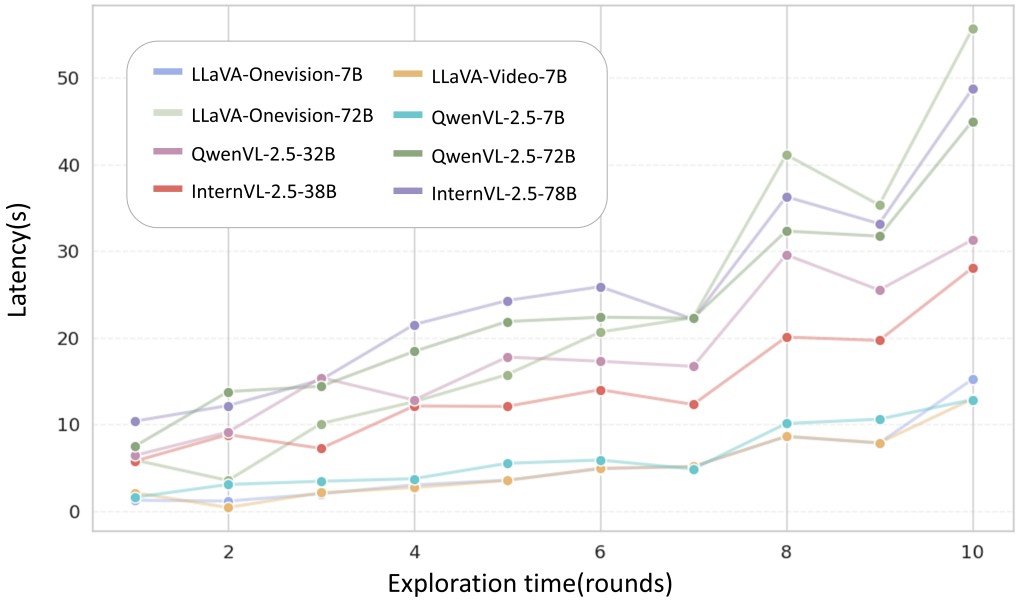

Figure 15: The trend of the model's inference time per question as the duration of exploration increases.

number of frames at every step. Thus, inference time scales approximately linearly with history length.

To provide context, we also measured human inference time. While average latency isn't directly comparable due to individual variation, we find that for all human testers, response time remained stable regardless of how long the exploration had lasted. This starkly contrasts with model behavior. The underlying reason is that humans can actively abstract and compress information throughout the exploration process, forming an internal knowledge base. Rather than treating each question as a fresh input, humans recall previously formed abstractions, enabling efficient reasoning without reprocessing all historical data.

This comparison highlights a critical need: for models to perform well in real-world embodied tasks, they must learn to dynamically distill and retain knowledge during exploration. Instead of passively accumulating history or answering questions in isolation, models should develop mechanisms to summarize and store essential information in an efficient, retrievable form, paving the way for scalable and real-time embodied reasoning.

# E  Social Impact

OST-Bench aims to advance the development of multimodal large language models (MLLMs) with stronger online spatio-temporal reasoning capabilities, which are critical for real-world embodied tasks such as assistive robotics, autonomous navigation, and human-robot interaction. By introducing a more realistic and challenging benchmark, we hope to drive progress toward more reliable and generalizable agents capable of perceiving and reasoning in real-world environments under online settings. However, as the benchmark assumes a static environment and focuses only on perception and reasoning, there is a risk of overestimating model readiness for real deployment. Caution is needed to avoid misuse or overreliance on models without broader capabilities like interaction or manipulation, which are essential for safe and responsible AI integration in the real world.

# F License and Acess

## F.1 License and Acess for Existing Assets

As mentioned in the main paper, our real-world scene data is sourced from ScanNet, Matterport3D, and ARKitScenes. To access and use these three datasets, users should follow their original licenses [4, 3, 1], and ask their official hosts for authorization. Additionally, our annotated data come from EmbodiedScan and MMScan, access to these datasets requires submitting a request via a Google Form [2] and following the license attached to the form.

We use ScanNet, Matterport3D, and ARKitScenes as the scene data and leverage the video information provided in them. We adopt the bounding box annotations and textual annotations from EmbodiedScan and MMScan as the base datasets for our benchmark. Throughout the usage of these datasets, their licenses and terms of use are properly respected.

## F.2 License and Acess for OST-Bench

The OST-Bench dataset is distributed under the Creative Commons Attribution 4.0 International License (CC BY 4.0) and available for direct download at `https://github.com/rbler1234/OST-Bench` or `https://www.kaggle.com/datasets/jinglilin/ost-bench/data`.

We release our benchmark under the CC-BY license and Terms of Use, and require that any use of the dataset for model evaluation be properly disclosed. This license supplements but does not override the original licenses of source materials; users must also comply with all relevant legal requirements concerning data subjects. This statement clarifies the obligations and liabilities associated with using this benchmark. While we strive to ensure the accuracy and legality of all samples, we do not guarantee their absolute completeness or correctness. We assume no responsibility for any legal or other issues that may arise from the use of OST-Bench, including but not limited to copyright infringement, privacy violations, or the misuse of sensitive information. By accessing, downloading, or using OST-Bench, you acknowledge that you accept this statement and agree to comply with the full terms of the CC-BY license. If you do not agree with these terms or the CC-BY license, you are not permitted to use this benchmark. OST-Bench will be hosted and maintained on GitHub and the Kaggle platforms.

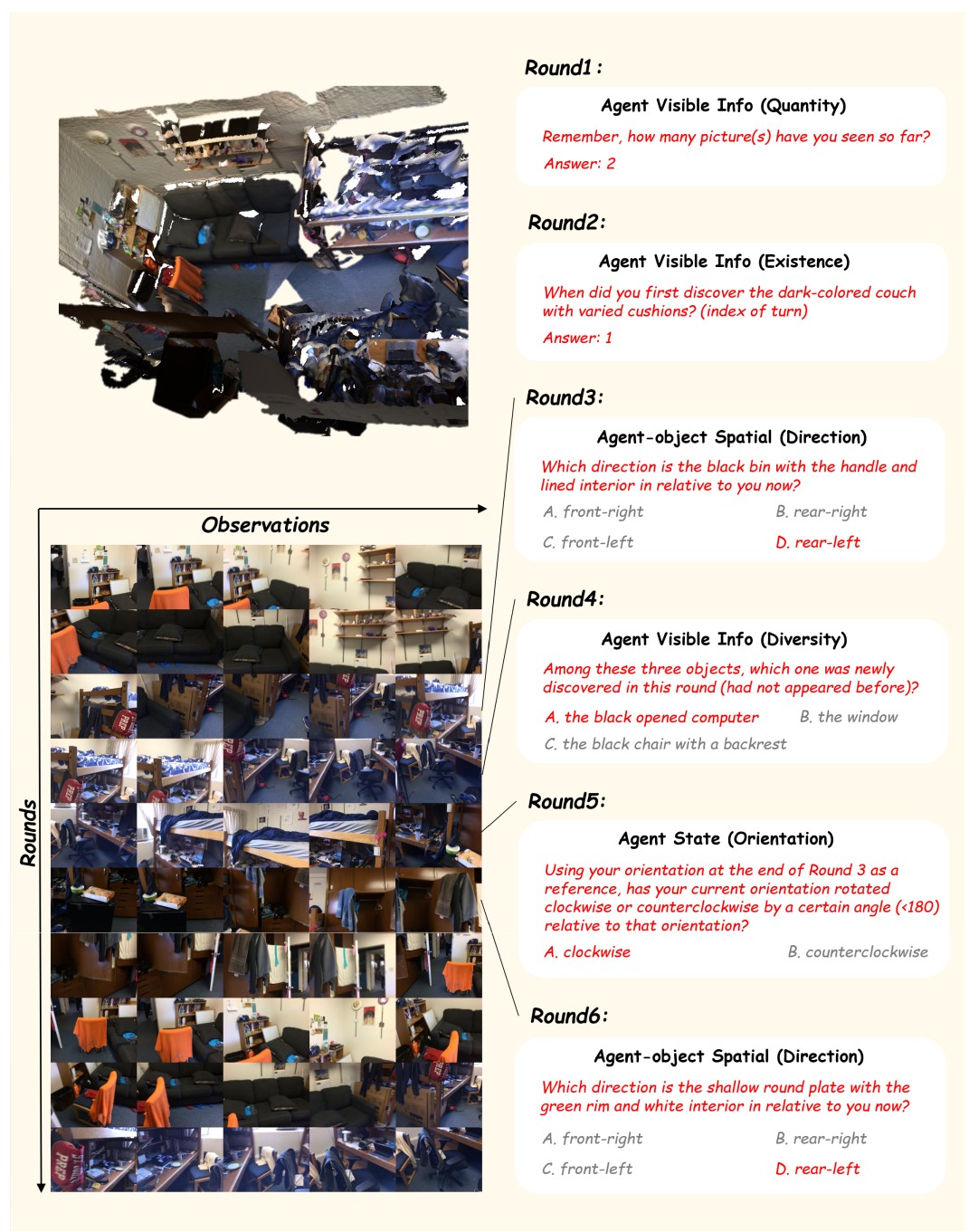

**Round1:**

**Agent Visible Info (Quantity)**

*Remember, how many picture(s) have you seen so far?*

*Answer: 2*

**Round2:**

**Agent Visible Info (Existence)**

*When did you first discover the dark-colored couch with varied cushions? (index of turn)*

*Answer: 1*

**Round3:**

**Agent-object Spatial (Direction)**

*Which direction is the black bin with the handle and lined interior in relative to you now?*

*A. front-right*          *B. rear-right*

*C. front-left*          *D. rear-left*

**Round4:**

**Agent Visible Info (Diversity)**

*Among these three objects, which one was newly discovered in this round (had not appeared before)?*

*A. the black opened computer*          *B. the window*

*C. the black chair with a backrest*

**Round5:**

**Agent State (Orientation)**

*Using your orientation at the end of Round 3 as a reference, has your current orientation rotated clockwise or counterclockwise by a certain angle (<180) relative to that orientation?*

*A. clockwise*          *B. counterclockwise*

**Round6:**

**Agent-object Spatial (Direction)**

*Which direction is the shallow round plate with the green rim and white interior in relative to you now?*

*A. front-right*          *B. rear-right*

*C. front-left*          *D. rear-left*

Figure 16: **Example 1 of OST-Bench data samples.** Each row represents the newly added observations in each round, with images input from left to right within each round. The example shows the question-answer pairs from the first six rounds.

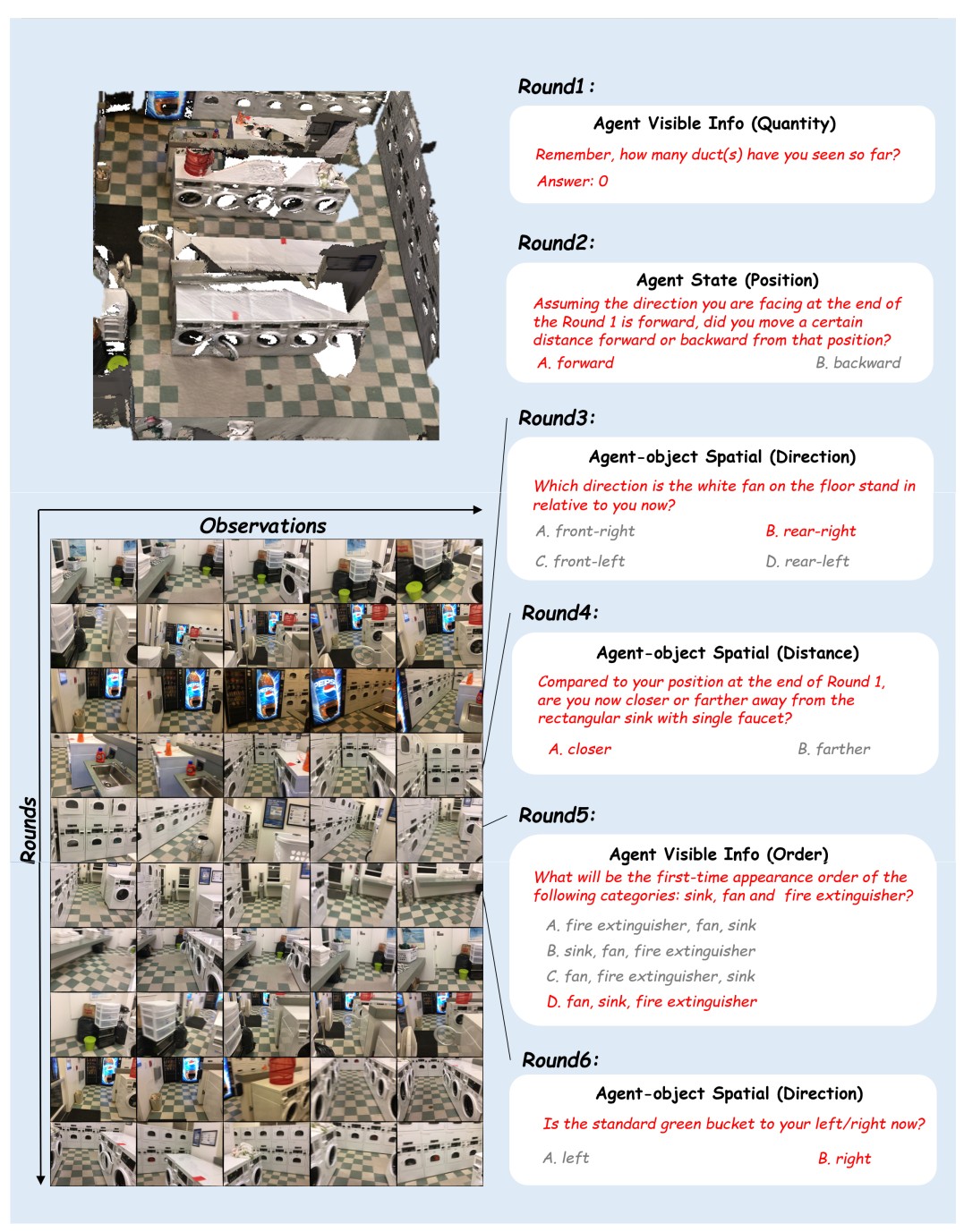

Figure 17: **Example 2 of OST-Bench data samples.** Each row represents the newly added observations in each round, with images input from left to right within each round. The example shows the question-answer pairs from the first six rounds.

