# OpenReview forum: "OST-Bench: Evaluating the Capabilities of MLLMs in Online Spatio-temporal Scene Understanding"
_NeurIPS.cc/2025/Datasets_and_Benchmarks_Track — NeurIPS 2025 Datasets and Benchmarks Track poster_

### Official Review · Reviewer_KAnX · 2025-06-28

**Rating:** 5
**Confidence:** 3

**Summary:**

The paper presents OST-Bench, a new benchmark for assessing the spatio-temporal reasoning abilities of multimodal large language models (MLLMs) in dynamic, online settings. It simulates an embodied agent exploring 3D environments over time, requiring models to integrate sequential visual observations for complex reasoning. Experiments reveal that while MLLMs handle object visibility well, they perform poorly on spatial relationships and long-term memory tasks, with accuracy dropping significantly as exploration progresses. OST-Bench exposes key limitations in current models and offers a challenging benchmark for advancing embodied scene understanding.

**Dataset Code Accessibility:**

Yes

**Ethical Considerations:**

No, there are no or only very minor ethics concerns

**Final Justification:**

The author's rebuttal has resolved my concerns, and I maintain my original rating.

**Limitations Weaknesses:**

- The ablation analysis is relatively coarse and lacks a deeper investigation into how different factors—such as task type, question length, spatial complexity, and object occlusion—affect model performance. It is recommended that the authors conduct a more fine-grained analysis of the benchmark's key challenges.

- While the benchmark focuses on complex spatio-temporal reasoning, all evaluations are conducted in a zero-shot setting without incorporating prompting strategies such as Chain-of-Thought (CoT). It is suggested that the authors explore integrating CoT into the evaluation process to better assess the reasoning capabilities of MLLMs.

**Strengths Contributions:**

- OST-Bench is specifically designed for online spatio-temporal scene understanding, covering three core tasks: agent state estimation, visible object recognition, and agent-object spatial reasoning. It fills a critical gap in evaluating dynamic reasoning and memory capabilities of multimodal models.

- By introducing frame-by-frame exploration and multi-round QA settings, OST-Bench effectively simulates real-world embodied perception, enabling a unique assessment of long-term memory and multi-step spatial inference in large models.

- Evaluations of both proprietary and open-source MLLMs, along with error type analysis and cross-frame reasoning studies, reveal significant limitations in memory integration and spatial anchoring—highlighting key directions for future model development.

---

> ### Author Rebuttal · Authors · 2025-07-29
>
> Thank you for recognizing the value of our work on online spatio-temporal reasoning and for your thoughtful and encouraging feedback. We appreciate your recognition of OST-Bench as a meaningful benchmark for evaluating dynamic memory and spatial reasoning in MLLMs, as well as your suggestions regarding the coarse ablation analysis and the potential value of incorporating Chain-of-Thought (CoT) prompting. Below are our detailed answers.
>
> ## Q1: More Detailed Ablation Study
> In the main paper , we analyze the impact of task type and exploration time on model performance. Below, we provide additional fine-grained analysis covering question length and spatial complexity. As all objects referenced in our benchmark are ensured to be visible in the agent’s past observations (Sec. 3.2, Appx. A.2), object occlusion is not a confounding factor. To isolate the effect of each factor, we report the average scores across all models for each analysis below. (In all tables, “JUD.”, “CNT.”, “TEMP.”, and “EST.” are abbreviations for “judgment”, “counting”, “temporal localization”, and “estimation”, respectively. “A. State”, “A. Info”, and “AO” refer to “Agent State”, “Agent Visible Info”, and “Agent-Object Spatial Relationship”.)
> ### A. Effect of Task Type
> | Format/Type |  JUD. | TEMP. |  CNT. | EST. |   | A. State | A. Info |   AO.  |
> |:-----------:|:-----:|:---------:|:-----:|:-----:|:-:|:-------:|:------:|:-----:|
> | Model(avg.) | 57.60 |   42.24   | 59.44 | 26.20 |   |  41.90  |  65.08 | 33.53 |
> These are some key findings:
>
> - Models perform significantly better on the *Agent Visible Info* task compared to other categories. This suggests that current models are capable of dynamically perceiving scene information with temporal awareness, but struggle with more complex spatio-temporal reasoning tasks.
>
> - Models perform particularly poorly on Estimation tasks, highlighting their weak numerical reasoning and the inherent difficulty of this question format.
>
> - Additional findings (see Appendix C.4): Models show better understanding of distance than direction; models are relatively good at detection, but poor at counting.
>
> ### B. Effect of Exploration Time
>    As illustrated in the main paper(Fig. 3), we observe a significant decline in  model accuracy as the agent continues to explore with an increasing number of sequential observations  in the online setting.
>
> ### C. Effect of Question Length
> |Length |  0-50 | 50-100 | 100-150 | 150-200 | 200-250 | 250-300 |
> |:------:|:-----:|:------:|:-------:|:-------:|:-------:|:-------:|
> | Model(avg.) | 83.01 |  48.95 |  40.25  |  39.30  |  44.93  |  71.18  |
>
>   We observe a U-shaped trend, where models perform better on extremely short and extremely long questions. Upon investigation, we find this is largely due to task-type distribution: 87% of questions in the 0–50 range are Agent Visible Info Existence subtype, over 95% in the 250–300 range are Agent Visible Info Diversity subtype— both of which are subtypes where models generally achieve high accuracy.
>   Thus, the apparent correlation between question length and performance is largely a byproduct of task-type distribution, rather than complexity due to length per se.
>
> ### D. Effect of Spatial Complexity
>   We use the quantity of objects in the observation as a proxy for spatial complexity:
> | spatial complexity (object quantity) |  0-15 | 15-30 | 30-45 | 45-60 | 60-75 | 75-90 | 90-105 |
> |:-----------------------------:|:-----:|:-----:|:-----:|:-----:|:-----:|:-----:|:------:|
> |          Model(avg.)          | 55.29 | 41.54 | 39.36 | 36.89 | 37.87 | 38.87 |  33.56 |
>
>   We observe a clear downward trend in model accuracy as spatial complexity increases. This is expected—more objects lead to more complex spatial relations, making the reasoning task harder for models.
>
> ## Q2: CoT Experiments
>
> Since our benchmark targets complex spatio-temporal reasoning, we have explored different prompting strategies to better evaluate model capabilities. Specifically, we tested the following three settings:
> - **Answer-only:** The model is prompted to output only the final answer.
> - **Answer + Reasoning (default setting):** The model is asked to output both the answer and a justification using the prompt:
> *“Give me your answer and your reason in a JSON format.”*
>   This format encourages explicit reasoning and also facilitates manual error analysis.
> -  **Chain-of-Thought (CoT):**  We prompt the model to follow a structured reasoning process to think step by step:
>    ```
>     <Step 1>: Planning – understand the question and identify necessary information.
>     <Step 2>: Collecting – summarize relevant info from each dialogue turn.
>     <Step 3>: Reasoning – integrate collected information to derive the answer.
>    ```
> We evaluated three representative models—InternVL2.5-8B, InternVL2.5-38B, and GPT-4o—under all three settings. (In the  table, “JUD.”, “CNT.”, “TEMP.”, and “EST.” are abbreviations for “judgment”, “counting”, “temporal localization”, and “estimation”, respectively. “A. State”, “A. Info”, and “AO” refer to “Agent State”, “Agent Visible Info”, and “Agent-Object Spatial Relationship”.) Surprisingly, CoT did not significantly improve overall performance; across different task types and question formats, it did not lead to consistent gains across models, and in some cases, even resulted in performance drops for certain question types. This finding is consistent with prior works (e.g., VSI[1]) that, while CoT helps in pure-text reasoning, its benefits in multimodal spatial reasoning remain limited.
>
> [1] Thinking in Space: How Multimodal Large Language Models See, Remember and Recall Spaces (CVPR 2025)
>
> |          Method          | **Overall** | JUD. | EST. | CNT. | TEMP. | A. State | A. Info |   AO.  |
> |:------------------------:|:-----------:|:----:|:----:|:----:|:---------:|:-------:|:------:|:----:|
> |      GPT-4o (direct)     |   **50.8**  | 60.5 | 32.9 | 53.5 |    49.2   |   44.4  |  70.5  | 37.7 |
> |     GPT-4o  (reason)     |   **49.5**  | 59.8 | 25.2 | 61.2 |    50.2   |   39.9  |  72.7  | 34.6 |
> |       GPT-4o  (CoT)      |   **50.2**  | 61.2 | 26.1 | 57.5 |    52.4   |   40.5  |  71.9  | 37.6 |
> |  InternVL2.5-8B (direct) |   **44.5**  | 52.4 | 30.4 | 57.1 |    39.5   |   44.5  |  59.2  | 31.7 |
> |  InternVL2.5-8B (reason) |   **44.6**  | 52.4 | 29.3 | 56.4 |    42.7   |   41.2  |  58.6  | 34.6 |
> |   InternVL2.5-8B (CoT)   |   **43.8**  | 52.2 | 29.0 | 52.1 |    42.6   |   41.8  |  57.1  | 34.7 |
> | InternVL2.5-38B (direct) |   **50.4**  | 61.4 | 32.3 | 56.5 |    45.7   |   44.4  |  72.4  | 35.5 |
> | InternVL2.5-38B (reason) |   **50.8**  | 61.2 | 31.8 | 60.3 |    45.9   |   45.5  |  73.8  | 34.2 |
> |   InternVL2.5-38B (CoT)  |   **50.8**  | 60.7 | 33.2 | 59.7 |    46.5   |   45.5  |  73.4  | 34.6 |
>
> *Note*: Compared to the submitted version, we conducted a thorough manual review of the dataset and corrected a very small portion of erroneous data (less than 5%). We also refined the answer extraction process from model outputs. As a result, the reported model results may differ slightly from those in the main paper, but these changes do not affect any of the paper’s main conclusions.

---

> > ### Author Response · Authors · 2025-08-07
> >
> > Dear Reviewer,
> >
> > Thank you again for your time and constructive feedback. We’ve now submitted our response and would be grateful to hear your thoughts—particularly whether your earlier concerns have been addressed to your satisfaction.
> >
> > Best regards,
> >
> > The Authors

---

> > ### Comment · Reviewer_KAnX · 2025-08-08
> >
> > Thanks for your response. The author's rebuttal has resolved my concerns, and I maintain my original rating.

---

### Official Review · Reviewer_nqKx · 2025-07-01

**Rating:** 4
**Confidence:** 3

**Summary:**

This paper introduces OST-Bench, a novel benchmark designed to evaluate the online spatio-temporal reasoning capabilities of multi-modal large language models (MLLMs). OST-Bench emphasizes the dynamic understanding of scenes from an embodied agent's perspective, focusing on online processing and spatio-temporal awareness. The benchmark consists of 1.4k scenes and 10k question-answer pairs collected from ScanNet, Matterport3D, and ARKitScenes. The authors evaluate several leading MLLMs on OST-Bench and find that they struggle with complex spatio-temporal reasoning tasks, especially as the exploration horizon extends and memory grows. The paper highlights the need for advancements in online embodied reasoning and provides insights into the core challenges that must be addressed to improve model performance.

**Dataset Code Accessibility:**

Yes

**Ethical Considerations:**

No, there are no or only very minor ethics concerns

**Limitations Weaknesses:**

1. The benchmark assumes a static environment where object positions and states remain unchanged. This limitation does not reflect real-world scenarios where objects and environments can change dynamically.
2.  OST-Bench focuses solely on the agent's online perception and reasoning abilities, excluding other crucial embodied task capabilities such as interactive behaviors and active manipulation. This narrow focus may not fully capture the complexity of real-world embodied tasks.
3. There is a significant performance gap between the best MLLMs and human performance, indicating that current models are not yet ready for real-world applications requiring robust online spatio-temporal reasoning.

**Strengths Contributions:**

1. OST-Bench is the first benchmark to focus on online spatio-temporal understanding from an embodied agent's perspective. This unique approach better reflects real-world embodied perception and reasoning challenges.
2. The benchmark covers three main task categories (Agent State, Agent Visible Info, and Agent-Object Spatial Relationship) with 15 fine-grained question subtypes, providing a thorough evaluation of MLLMs' capabilities in dynamic scene understanding.
3. Built on high-quality datasets like ScanNet, Matterport3D, and ARKitScenes, OST-Bench ensures that the evaluation is grounded in real-world scenarios, enhancing the relevance and applicability of the benchmark.

---

> ### Author Rebuttal · Authors · 2025-07-29
>
> Thank you for recognizing OST-Bench’s contributions to online spatio-temporal reasoning, and for your constructive feedback on its lack of dynamic setting, limited interactivity, and the model-human performance gap.
> ## Q1 & Q2: Lack of Dynamic Setting & Limited Interactivity
>  Thank you for raising this important concern. We acknowledge this as a limitation of our current work and offer the following clarifications:
>
> - To better reflect real-world performance, our benchmark is built on real-world environments rather than simulation, aiming to reduce the sim-to-real gap. However, most available real-world datasets inherently feature static scenes, which aligns with our current focus.
> - As shown in our results, even without dynamic object movement or interaction, online spatio-temporal reasoning remains a significant challenge for current models. We believe that reasoning in static scenes is a foundational capability—models must first be competent in this setting before tackling more complex, dynamic tasks.
> - That said, we fully agree that dynamics and interactivity are important components of embodied reasoning. Extending OST-Bench to include dynamic and interactive environments is a key direction of our future work to make the benchmark more comprehensive.
>
> ## Q3: Model-Human Performance Gap
>  Our results clearly show that current MLLMs still fall short of the performance required for real-world applications involving robust online spatio-temporal reasoning. To better understand these limitations, we conducted a detailed analysis in Section 4.3 of the main paper, identifying two core challenges and corresponding directions for improvement:
> - (A) Enhancing the model's ability to retrieve key information from long-term, temporally extended memory;
> - (B) Improving the integration of spatial cues for complex spatial reasoning.
>
>  Furthermore, we find that training with sufficient high-quality spatial reasoning data leads to notable performance gains on OST-Bench. Below are results from two fine-tuning settings (In all tables, “JUD.”, “CNT.”, “TEMP.”, and “EST.” are abbreviations for “judgment”, “counting”, “temporal localization”, and “estimation”, respectively. “A. State”, “A. Info”, and “AO” refer to “Agent State”, “Agent Visible Info”, and “Agent-Object Spatial Relationship”):
>
> - **In-domain SFT** (using 50k training samples generated from OST-Bench’s training split of 7k scenes):
>
> | Method          | Overall | JUD.  | EST.  | CNT.  | Temp-Loc. | A State | A Info | AO    |
> |-----------------|---------|-------|-------|-------|-----------|---------|--------|-------|
> | QwenVL2.5-7B    | 41.16   | 50.13 | 22.73 | 62.10 | 36.6      | 40.43   | 52.56  | 31.53 |
> | + Fine-tuned    | 54.03   | 58.99 | 41.21 | 74.63 | 50.20     | 48.25   | 69.82  | 43.48 |
> | InternVL2.5-8B  | 44.62   | 52.38 | 29.28 | 56.40 | 49.2      | 41.87   | 58.60  | 34.61 |
> | + Fine-tuned    | 57.41   | 64.11 | 38.45 | 74.92 | 57.47     | 43.99   | 79.28  | 46.26 |
> | InternVL2.5-38B | 50.78   | 61.39 | 31.50 | 61.10 | 45.93     | 45.38   | 73.88  | 33.95 |
> | + Fine-tuned    | 60.16   | 68.39 | 44.09 | 73.12 | 56.09     | 50.77   | 81.74  | 47.45 |
>
> - **Out-of-domain**  (fine-tuning on external spatial reasoning datasets like MMScan, RoboPoint, RoboVQA) also leads to similar improvements.
>
> | Method                       | Overall | JUD.  | EST.  | CNT.  | TEMP. | A. State | A. Info | AO.    |
> |------------------------------|---------|-------|-------|-------|-----------|---------|--------|-------|
> | InternVL3-8B                 | 44.96   | 54.03 | 30.58 | 60.32 | 37.30     | 45.84   | 56.82  | 32.51 |
> | + Fine-tuned (out-of-domain) | 48.50   | 59.32 | 29.53 | 69.21 | 29.53     | 40.34   | 66.34  | 36.07 |
>
> We believe that enhancing retrieval from long-term memory, improving complex spatial reasoning capabilities, and fine-tuning with larger and higher-quality spatial reasoning data are three promising paths toward addressing the challenges of real-world online reasoning.
>
> *Note*: Compared to the submitted version, we conducted a thorough manual review of the dataset and corrected a very small portion of erroneous data (less than 5%). We also refined the answer extraction process from model outputs. As a result, the reported model results may differ slightly from those in the main paper, but these changes do not affect any of the paper’s main conclusions.

---

> > ### Author Response · Authors · 2025-08-07
> >
> > Dear Reviewer,
> >
> > Thank you again for your constructive feedback. We have now submitted our response and hope it provides clarity on the points you raised. We would greatly appreciate knowing whether your concerns have been adequately addressed.
> >
> > Best regards,
> >
> > The Authors

---

### Official Review · Reviewer_fZPw · 2025-07-03

**Rating:** 5
**Confidence:** 4

**Summary:**

The paper proposes OST-Bench, an MLLM evaluation benchmark where a sequence of frame trajectories are provided for an MLLM simulating an embodied agent that turns and moves in a 3D environment. The task is to see if MLLMs are capable of online spatiotemporal reasoning, which sometimes requires recalling information from previous turns' observations in a multi-turn conversation format. The dataset is generated by using 1.4k samples from test and validation sets of ScanNet, ARKitScenes, and Matterport3D and leveraging corresponding annotations in EmbodiedScan and MMScan to create the multi-turn exploration routes and questions in each turn. For questions, the authors use a rule-based generation technique to inject annotations into question templates. The questions are categorized into three categories and seven subcategories based on type of the answer and reasoning required to answer it. Multiple proprietary and open-source models are evaluated compared on these trajectories. The shortcomings of VLMs in this type of online spatiotemporal reasoning is highlighted by comparison to human and chance performance. Specifically, authors show longer sequences that may require recalling and reasoning on past turns are major failure modes. They further characterise VLM sources of error by manual inspection.

**Additional Feedback:**

1. In table 2, for each section and column, bold the best and underline the second-best performance

2. line 62 type: both dimensions

3. line 42 type: To more accurately simulate ... OST-Bench defines tasks

**Dataset Code Accessibility:**

Partly

**Dataset Code Comments:**

Both the github repo and Kaggle page are well-organized and well-documented. All evaluation data is available. Code to run all VLM evals are available in the repo. No train split is provided. The code to generate the dataset from parent datasets and annotations is not provided.

**Ethical Comments:**

There is no information about how human evaluators were recruited and compensated (your human performance baselines). Please specify the details and include them in the appendix.

**Ethical Considerations:**

Yes, there are ethics concerns that require attention by the authors

**Final Justification:**

The authors have addressed my main concern regarding availability of a train split and also provide new results with open-source VLMs fine-tuned on this split. They also promise to release the data generation code.

The authors also addressed my concerns regarding ambiguity of quality control protocol and annotator compensation.

Conditioned on the release of training split and data generation code, I am increasing my score.

**Limitations Weaknesses:**

1. My main concern is the lack of a train split for the dataset. Even if you don't have the compute to fine-tune open-source VLMs and measure and report generalization on the test set, this train split could be used by the community for such purposes. Given your automatic data generation pipeline, I assume creating a train set should not be hard.

2. The code to generate the dataset and questions from the base datasets and annotations is not provided.

3. The manual data quality check (line 178) is vague. Describe the exact protocol for quality check and exact criteria for removing samples.

**Strengths Contributions:**

1. As far as I know, this benchmark is the first to create this kind of multi-turn visual question answering, i.e., online egocentric spatio-temporal reasoning and in this sense is novel and complements existing 3D reasoning benchmarks (e.g. the ones in Table 1). The question categories and subtypes cover a diverse set of spatiotemporal reasoning tasks with respect to agent and/or surrounding objects.

2. Automatic trajectory generation and rule-based questions make it possible to scale the dataset without the need for human annotators.

3. The evaluation is done using both proprietary and open-source models, and the shortcomings and failure modes e.g. regarding long-term recall reasoning, that cause a large gap between VLM and human performance are discussed.

4. The quality of writing is acceptable. Although some paragraphs are a bit verbose. A further polish is recommended.

5. Both the github repo and Kaggle page are well-organized and well-documented. Evaluation data is available. Code to run all VLM evals are available in the repo.

---

> ### Author Rebuttal · Authors · 2025-07-29
>
> Thank you for recognizing the novelty and value of our work. We also appreciate your suggestions regarding the release of our code and data, as well as your constructive comments on writing. We will follow your recommendations to improve the openness of our project and further polish the paper accordingly.
> ## Q1: The Train Split of OST-Bench
> Yes, given our automatic data generation pipeline, generating training data is indeed straightforward. In fact, the training and test splits of OST-Bench were generated simultaneously. As our current submission focuses on benchmark evaluation, we only released the test split.
>   Specifically, our test set includes 10k samples from 1.4k test scenes drawn from ScanNet, Matterport3D, and ARKitScenes. Correspondingly, our training split contains **50k** samples from **7k** training scenes across the same datasets. We have internally fine-tuned VLMs on this in-domain training set, and the results show a substantial performance boost on OST-Bench (In the table, “JUD.”, “CNT.”, “TEMP.”, and “EST.” are abbreviations for “judgment”, “counting”, “temporal localization”, and “estimation”, respectively. “A. State”, “A. Info”, and “AO” refer to “Agent State”, “Agent Visible Info”, and “Agent-Object Spatial Relationship”):
>
> | Method          | Overall | JUD.  | EST.  | CNT.  | Temp-Loc. | A State | A Info | AO    |
> |-----------------|---------|-------|-------|-------|-----------|---------|--------|-------|
> | QwenVL2.5-7B    | 41.16   | 50.13 | 22.73 | 62.10 | 36.6      | 40.43   | 52.56  | 31.53 |
> | + Fine-tuned    | 54.03   | 58.99 | 41.21 | 74.63 | 50.20     | 48.25   | 69.82  | 43.48 |
> | InternVL2.5-8B  | 44.62   | 52.38 | 29.28 | 56.40 | 49.2      | 41.87   | 58.60  | 34.61 |
> | + Fine-tuned    | 57.41   | 64.11 | 38.45 | 74.92 | 57.47     | 43.99   | 79.28  | 46.26 |
> | InternVL2.5-38B | 50.78   | 61.39 | 31.50 | 61.10 | 45.93     | 45.38   | 73.88  | 33.95 |
> | + Fine-tuned    | 60.16   | 68.39 | 44.09 | 73.12 | 56.09     | 50.77   | 81.74  | 47.45 |
>
>   We would be glad to release the training split of OST-Bench to facilitate future research. However, as NeurIPS policy prohibits introducing new links or updating code/dataset repositories during the rebuttal phase, we will release the OST-Bench training set immediately after the rebuttal period. Please stay tuned!
>
> ## Q2: The Data Generation Code
>
> Our data generation process follows the description provided in Supplementary Section A. As per NeurIPS rebuttal policy, we are not allowed to introduce new links or update code and dataset repositories during the rebuttal phase. However, we will release the code for generating the dataset and questions from the base datasets and annotations immediately after the rebuttal period.
>
> ## Q3:  Detailed Description of the Quality Control Protocol
>
> Below we provide the exact protocol and criteria used for manual quality checking. During quality inspection and data cleaning, each sample is evaluated for validity based on the following criteria:
> ###   Question Validity:
> - **Object visibility**: All objects mentioned in the question must be clearly visible in the agent's past observations.
> (e.g. For the question "Is the white pillow to your left/right?", if no white pillow appears in any previous observation, the sample is considered invalid.)
> - **Object referential clarity**: Any description of objects must uniquely identify a single object in the scene, avoiding ambiguity.(e.g. For the question "What is the horizontal distance between you and the sofa?", if multiple sofas are present and indistinguishable, the sample is invalid.)
>  - **Answerability by humans**: The question should be answerable by a human with reasonable confidence.(e.g.
>  (1) "Which one is closer to you, Object A or Object B?" is invalid if A and B are at nearly equal distances, making comparison ambiguous.
>  (2) "How many chairs have you seen so far?" is invalid if, for instance, a furry object is hard to distinguish as a chair or a sofa.)
> ###   Answer Accuracy:
> - If the question is valid, the correctness of the provided answer is verified. Incorrect answers lead to sample invalidity.
>
> Before submission, we randomly sampled and manually reviewed a subset of the data following this protocol, ensuring the sample error rate remained below 5%. Subsequently, we conducted a comprehensive manual review of all 10k test samples, correcting any issues and guaranteeing “100%” data accuracy under our protocol.
>
> ## Q4: Ethical Considerations Information
>
> Thank you for pointing out this important aspect. We will include a dedicated section on ethical considerations in the final version of the main paper and supplementary materials as suggested. For our human performance baselines, we recruited 10 annotators with diverse backgrounds via social media and compensated each with $50 for their participation.
>
> ## Q5: A Recommended Further Polish of the Paper
>
> We sincerely appreciate your careful reading and constructive feedback. We will revise Table 2 to improve clarity and fix the two typos you identified. In addition, we will polish the writing throughout the paper, particularly in paragraphs that are currently too verbose, to enhance readability and conciseness in the final version.

---

> > ### Comment · Reviewer_fZPw · 2025-08-01
> >
> > Thank you for your response.
> >
> > The authors have addressed my main concern regarding availability of a train split and also provide new results with open-source VLMs fine-tuned on this split. They also promise to release the data generation code (I believe you can put these on your Kaggle page and Github repo now without violating conference policy--you don't need to share a link here. Just let the reviewers know and we can check them).
> >
> > The authors also addressed my concerns regarding ambiguity of quality control protocol and annotator compensation.
> >
> > Conditioned on the release of training split and data generation code, I am increasing my score.

---

> > > ### Author Response · Authors · 2025-08-02
> > >
> > > We sincerely appreciate your thoughtful feedback and constructive suggestions throughout the review process. Your insights have been invaluable in helping us refine and improve our work. We are truly grateful for your recognition of our efforts to address the key concerns raised.
> > >
> > > Following your guidance, while respecting the policy by not including direct links here, we have now updated both our GitHub repository and Kaggle page with:
> > > - The complete training split annotations.
> > > - The full data generation pipeline code.
> > >
> > > You can access the updates via the original submission links and check them. Thank you once again for your time and consideration.

---

### Official Review · Reviewer_uedK · 2025-07-07

**Rating:** 5
**Confidence:** 4

**Summary:**

The paper presents a benchmark for evaluating the online spatio-temporal understanding of large multimodal models (MLLMs) – going away from current benchmarks that evaluate the agent's understanding in an offline setting with pre-recorded inputs. Their idea is to dynamically and incrementally keep probing the agent's understanding every few frames throughout its trajectory – by processing incrementally acquired visual observations and integrate them with historical memory to answer questions about their own state, the visible scene, and spatial relationships to objects in the scene. Towards this end, they contribute a dataset with 10k QA pairs across 1.4k scenes across multiple popular datasets and evaluations of several proprietary and open-source MLLMs on this benchmark. Their findings and comprehensive failure analysis suggests that current state-of-the-art MLLMs are significantly lag behind human performance – especially for longer sequences – predominantly due to reasoning-based errors.

**Additional Feedback:**

- Types of errors: I felt the types of errors identified in section 4.3.1 were more reflecting the question types as opposed to the agent's behavior. Some questions are more reasoning-based  (e.g. "how far is your position from where you were at round 2?") and others are more perception-based (e.g. "have you ever seen a bin?"). Therefore, it doesn't seem surprising that "agent state" errors had more reasoning errors and "agent visible info" questions had more perception errors.
- Would be nice to include plots that showcase the diversity of the types of scenes across the scene datasets used in the benchmark.

**Dataset Code Accessibility:**

Yes

**Dataset Code Comments:**

More information about the dataset and its availability is shared in the website they have mentioned in the paper:

https://github.com/rbler1234/OST-Bench

This includes Kaggle and Huggingface links. I downloaded the full dataset zip file from Kaggle and it looks pretty complete.

**Ethical Considerations:**

No, there are no or only very minor ethics concerns

**Final Justification:**

I am convinced with the authors' response to all my concerns in the rebuttal and believe this paper is a strong contribution to the community.

**Limitations Weaknesses:**

- My major concern with this paper is that the paper only presents results for general-purpose MLLMs. Regardless of the scale of data these models have been trained on, it has been shown time and again that they lack some sort of real-world grounding that is necessary for embodied tasks. Therefore, it is not surprising that these models are underperforming and unable to effectively utilize the memory and context to be able to answer questions correctly. I would be more interested in evaluating large-scale models that ground their understanding of the environment in some sort of memory representations (e.g. explicit semantic or topological maps or latent vectors)? These kinds of representations would make it significantly easier to answer a lot of the kinds of questions we see in this benchmark. These approaches might also be a little more interpretable – and would also give us a better sense of how far we are from human-level performance.
- Along similar lines, I am curious if the authors tried fine-tuning any of these models to understand how much of the performance gap can be recovered. I think that analysis would have been more useful than digging deeper into some error-cases that can somewhat be anticipated.
- I was going through the dataset and noticed that the image sequences seemed to have been captured at low FPSes (i.e. there was significant movement between consecutive frames). I am curious if the authors considered increasing the FPS – making it somewhat easier for the models with there being more overlap between consecutive frames.

**Strengths Contributions:**

- An agent's online spatio-temporal understanding of its presence in a novel scene is key for enabling it to succeed in most relevant tasks like object-goal navigation, rearrangement, social navigation, episodic question answering, and more. Therefore, a benchmark that evaluates agents on these capabilities seems like a useful stepping-stone to master before these agents can help people with their everyday tasks.
- The paper is nicely written and presented, flows well content-wise, and seems pretty complete with most lower-level questions addressed in the appendix.
- The dataset seems to be pretty valuable and I can appreciate the amount of engineering effort required to set up something like this with automated rule-based question generation across a wide variety of semantic categories – across multiple scene datasets.
The types of templated questions in the dataset seems pretty diverse
- The results and analysis section seemed pretty comprehensive with performance-vs-time plots, failure breakdowns with qualitative examples of "shortcut" behaviors, and cross-view question answering.

---

> ### Author Rebuttal · Authors · 2025-07-30
>
> Thank you for recognizing the value of our work and for providing valuable suggestions regarding additional evaluations, finetuning, and ablation studies. Below are our responses to each of your comments:
>
> ## Q1: Evaluating Large-scale Models with Memory Representations
>
> Following your suggestion, we evaluated several representative models that incorporate spatial grounding and memory representations to various extents, including Spatial-MLLM, VLM-3R, and LLaVA-3D.
> - Spatial-MLLM and VLM-3R follow a VGGT + VLM architecture and take RGB image sequences as input. In particular, VGGT provides a geometry-aware representation of the scene.
> - LLaVA-3D leverages RGB-D image sequences to encode 3D spatial information into 2D token representations.
>
> All three models were trained on spatial reasoning datasets and achieved strong performance on their respective benchmarks.
> We summarize their performance on OST-Bench below, comparing them to their corresponding base models (Spatial-MLLM[QwenVL2.5-3B]; VLM-3R and LLaVA-3D [LLaVA-Video-7B]):
>
> | Method                | Overall | JUD.  | EST.  | CNT.  | TEMP. | A. State | A. Info | AO.   |
> |-----------------------|---------|-------|-------|-------|-------|----------|---------|-------|
> | (base) QwenVL2.5-3B   | 34.82   | 47.92| 18.69| 59.37| 19.75 | 34.24| 47.54| 25.70|
> | Spatial-MLLM          | 26.82   | 37.30 | 21.88 | 29.52 | 15.31 | 25.47    | 39.38   | 20.90 |
> | (base) LLaVA-Video-7B | 39.28   | 52.77 | 16.13 | 63.10 | 33.80 | 33.50    | 58.32   | 28.80 |
> | VLM-3R                | 42.92   | 55.09 | 28.30 | 49.57 | 36.02 | 39.93    | 58.10   | 34.37 |
> | LLaVA-3D              | 30.06   | 46.10 | 5.9   | 13.45 | 36.26 | 29.7     | 38.4    | 26.30 |
>
> These are some key findings:
> - **Only VLM-3R shows overall improvement over its base model.** Both Spatial-MLLM and LLaVA-3D experience significant drops in performance. VLM-3R shows consistent improvement, particularly in Agent State, Agent-Object Spatial Relationship tasks, and questions that require estimation capabilities.
> - **Instruction-following capability degradation is evident**. Compared to their base models, which consistently follow prompts and output both answers and coherent reasoning, all these three models demonstrate poor adherence to our instruction format in their responses. Spatial-MLLM is limited to outputting floating-point numbers and multiple-choice options. All three grounded models frequently produce either no reasoning at all or reasoning that is incoherent or nonsensical.
> - **Poor generalization on out-of-domain tasks.** Despite strong performance on their training-aligned spatial benchmarks (e.g., VSI, MMScan), both Spatial-MLLM and LLaVA-3D fail to generalize well to OST-Bench, which includes diverse, complex spatio-temporal prompts and new task types. VLM-3R demonstrates a certain degree of generalization capability; however, its improvements remain limited.
>
>  We argue that while large-scale models with memory-based grounding can improve performance on tasks aligned with their training objectives, this often comes at the cost of generalization. These models may struggle with instruction following in out-of-distribution settings and underperform on previously simple tasks(such as Agent Visible Info tasks / counting questions in OST-Bench)—losing part of the base LLM’s generalization ability. They tend to excel only on in-domain tasks and fail to transfer effectively to datasets with different distributions. In contrast, OST-Bench features more diverse and complex prompts and task formats, which these models find difficult to handle.
> ## Q2: Fine-tuning models on OST-Bench
>
> Following your suggestion, we conducted fine-tuning experiments on QwenVL2.5-7B, InternVL2.5-8B, and InternVL2.5-38B. The training data were sourced from 7k training scenes in ScanNet, Matterport3D, and ARKitScenes, totaling 50k annotated samples. All models were fine-tuned for 1 epoch, and the results are as follows.
>
> | Method          | Overall | JUD.  | EST.  | CNT.  | Temp-Loc. | A State | A Info | AO    |
> |-----------------|---------|-------|-------|-------|-----------|---------|--------|-------|
> | QwenVL2.5-7B    | 41.16   | 50.13 | 22.73 | 62.10 | 36.6      | 40.43   | 52.56  | 31.53 |
> | + Fine-tuned    | 54.03   | 58.99 | 41.21 | 74.63 | 50.20     | 48.25   | 69.82  | 43.48 |
> | InternVL2.5-8B  | 44.62   | 52.38 | 29.28 | 56.40 | 49.2      | 41.87   | 58.60  | 34.61 |
> | + Fine-tuned    | 57.41   | 64.11 | 38.45 | 74.92 | 57.47     | 43.99   | 79.28  | 46.26 |
> | InternVL2.5-38B | 50.78   | 61.39 | 31.50 | 61.10 | 45.93     | 45.38   | 73.88  | 33.95 |
> | + Fine-tuned    | 60.16   | 68.39 | 44.09 | 73.12 | 56.09     | 50.77   | 81.74  | 47.45 |
>
> Overall, each model exhibited **a performance gain of over 10 %**, with the following key findings:
>   1. **Among the three major task categories**, the most significant improvement was observed in the Agent Visible Info tasks — particularly for models with smaller parameter sizes. While other two task categories also saw notable improvements, their accuracies remained nearly or below 50%, indicating that even after fine-tuning, models still struggle with tasks requiring complex spatio-temporal reasoning. This suggests that simple supervised fine-tuning on OST-Bench does not fully solve the benchmark’s core challenges.
>   2.  All four question formats benefited from fine-tuning, but deeper inspection of sample predictions revealed more nuanced observations:
>
>    - For estimation (EST) and judgement (JUD) tasks, although scores improved, this **did not reflect a meaningful enhancement** in the models' abilities. The models often predicted nearly identical values across different samples within the same subtype (such as always near 150 degree for angle prediction), suggesting memorization rather than true understanding; models tended to always predict the same option (e.g., always choosing “closer” over “farther”). While we balanced the dataset to avoid label bias during construction, slight imbalances (e.g., “closer” being marginally more frequent) may have been exploited by the models to gain performance through shortcuts rather than learning meaningful reasoning patterns.
>    - Furthermore, post-finetuning, models showed a loss in **instruction-following ability**. They often failed to produce both the final answer and the corresponding reasoning as instructed, making it impossible to supervise or assess the reasoning process.
>    - Therefore, we do not believe that performance gains on the same test set after training indicate a **true improvement** in spatio-temporal reasoning. Rather, these observations suggest that the models may be exploiting dataset-specific shortcuts or memorizing training distributions, without truly learning the underlying tasks.
>
> While fine-tuning significantly improves raw performance, a considerable gap remains compared to human-level accuracy. This highlights two key points: (1) data-only supervised fine-tuning is insufficient to solve the challenges posed by OST-Bench — improvements may also be required on the model architecture or training methodology side; and (2) the benchmark itself is both challenging and robust. Despite being constructed using templates, OST-Bench resists shortcut learning and cannot be easily exploited through superficial patterns in the training distribution.
>
> ## Q3: Model Performance After Increasing the FPS
> Following your suggestion, we conducted experiments to investigate the impact of increasing FPS on model performance in OST-Bench. Due to the unique data generation process of Matterport3D, it is challenging to increase FPS while keeping question samples consistent. Therefore, we conducted our FPS experiments on 8.5k samples from ScanNet and ARKitScenes, which together account for the majority of our benchmark and are sufficient for drawing conclusions.
>
> |FPS Ratio| GPT-4o | InternVL2.5-38B | QwenVL2.5-32B |
> |-----|--------|-----------------|---------------|
> | 1.0× | 49.67  | 50.36           | 46.24         |
> | 1.5× | 50.90  | 49.57           | 46.50         |
> | 2.0× | 48.07  | 50.47           | 45.99         |
>
>   Specifically, we increased the original FPS by 1.5× and 2× (keeping the question time and content fixed), and evaluated how model performance changed. We observed that increasing FPS led to no significant improvements—performance varied by less than 2%. We believe this is because our original frame selection process already ensured sufficient visual continuity by enforcing a minimum overlap threshold between frames (see Supplementary A.1), and during data checking, we manually ensured that the provided information was sufficient to answer the questions. Therefore, increasing the FPS did not introduce substantial new information for the models to exploit.
>
> ## Q4: Clarification on the Purpose and Interpretation of Error Type Analysis
> While Agent Visible Info tasks involve reasoning, it is generally simpler—such as temporal ordering or identity matching—compared to the more complex spatio-temporal reasoning required by Agent State tasks. Our error analysis aims to identify where models truly struggle: the prevalence of perception errors in Visible Info tasks suggests perception is the main bottleneck, while reasoning dominates the errors in Agent State tasks.
>
> Although this correlation between task and error type may seem intuitive, empirical validation is essential. As noted in the paper, reasoning errors account for over 60% of all failures, marking it as the primary bottleneck for current MLLMs on OST-Bench. This insight helps pinpoint the key limitations hindering model performance, offering guidance for future research.
>
> ## Q5: Nice to Include New Plots
> Thank you for the suggestion. We completely agree with your point, and we will include visualizations showcasing the diversity of scene types across the datasets used in the benchmark in the final version of our paper.

---

> > ### Comment · Reviewer_uedK · 2025-08-05
> >
> > I thank the authors for responding to all my concerns. I appreciate all the additional experiments with spatial grounding and memory representations, the finetuning experiments, and the impact of FPS. I believe including these into the manuscript will make the manuscript much more comprehensive, well-rounded, and insightful. I am going to update my final rating accordingly.

---

> > > ### Author Response · Authors · 2025-08-06
> > >
> > > We sincerely appreciate your positive feedback. We're glad to hear that the additional experiments with spatial grounding and memory representations, the finetuning experiments, and FPS analysis addressed your concerns and helped strengthen the manuscript. Thank you for your thoughtful evaluation and for updating your rating accordingly.

---

### Decision · Program_Chairs · 2025-09-18

**Decision:**

Accept (poster)

**Comment:**

This paper mainly focuses on evaluating online spatio-temporal understanding from the perspective of an agent actively exploring a scene. It
received three accept and one borderline accept. According to the feedback, most concerns were effectively addressed during the discussion. Its merits, including good writing, valuable dataset, comprehensive results and analysis, are well recognized. I agree with the reviewers and think this paper meets the requirements of this top conference.